# Cross-basin and cross-taxa patterns of marine community tropicalization and deborealization in warming European seas

Ocean warming and acidification, decreases in dissolved oxygen concentrations, and changes in primary production are causing an unprecedented global redistribution of marine life. The identification of underlying ecological processes underpinning marine species turnover, particularly the prevalence of increases of warm-water species or declines of cold-water species, has been recently debated in the context of ocean warming. Here, we track changes in the mean thermal affinity of marine communities across European seas by calculating the Community Temperature Index for 65 biodiversity time series collected over four decades and containing 1,817 species from different communities (zooplankton, coastal benthos, pelagic and demersal invertebrates and fish). We show that most communities and sites have clearly responded to ongoing ocean warming via abundance increases of warm-water species (tropicalization, 54%) and decreases of cold-water species (deborealization, 18%). Tropicalization dominated Atlantic sites compared to semi-enclosed basins such as the Mediterranean and Baltic Seas, probably due to physical barrier constraints to connectivity and species colonization. Semi-enclosed basins appeared to be particularly vulnerable to ocean warming, experiencing the fastest rates of warming and biodiversity loss through deborealization.

Global climate change is fundamentally altering life on Earth[1–3]. This alteration is primarily due to three universal ecological responses of species to global warming including[4] poleward distribution shifts, adjustments in phenology, and reductions in mean body size. Shifts in distribution or seasonal timing help species track their specific thermal niches[5–7], whilst body size reduction is a complex process involving physiological and/or eco-evolutionary responses[8–10]. A poleward distribution shift is the displacement (active or passive) of populations that track species' thermal preferences[11], which is mainly driven by the general trend of decreasing water temperature with latitude[12]. Although our knowledge of marine species responses to ocean warming in North Atlantic is substantial[13,14], comparative assessments across biotic groups and regions to identify the underlying ecological processes[15] associated with marine community expansion, retraction and dispersal constraints are limited to the uneven distribution of monitoring programmes[16–20]. The role of dispersal limitation and habitat fragmentation in community responsiveness to climate variation has been addressed more thoroughly in terrestrial groups such as plants[21,22] compared to marine biodiversity[20,23,24]. However, organisms' dispersal ability and ocean connectivity are also important in shaping marine species distribution[25,26]. For instance, the presence of the European landmass north of the Mediterranean basin poses a considerable limitation to the response of species to sea warming, as it constrains the poleward movement of adults or propagules to more thermally favourable conditions, especially jeopardising endemic species[27].

Physiological constraints limit the range of suitable environmental conditions for populations and species[28]. To maintain viable

✉e-mail: gchust@azti.es

populations under ocean warming, species might respond (actively or passively) by shifting latitudinally (generally poleward) or by changing their phenology (e.g. earlier onset of spawning, migration or return from dormancy)[11]. Latitudinal shifts will change the relative abundance of species at specific locations (Fig. 1). Long-term monitoring programmes based on permanent stations have detected such changes in North Atlantic fish communities associated with ocean warming[15], based on temporal changes of the Community Temperature Index (CTI). The CTI is a measure of the average thermal affinity of ecological communities weighted by the relative abundance of each species[29,30]. Thus, temporal change in CTI informs on the turnover of the relative abundance of species (*sensu* Vellend[31]) according to their thermal affinity and the response of the community to temperature change. Changes in the CTI can be decomposed into the four underlying ecological processes[15]. Positive CTI changes correspond to an increased prevalence in warm-affinity species (i.e., tropicalization), and/or a decrease in cold-affinity species (i.e., deborealization). Tropicalization is hence interlinked to leading (warm range) edges, whilst deborealization to trailing (cold range) edges (Fig. 1). Negative CTI changes correspond to an increase in cold-affinity species (i.e., borealization) and/or a decrease of warm-affinity species (i.e., detropicalization). The reason why tropicalization may prevail (or not) over deborealization has been recently debated[32–34]. Assuming an ecophysiological equilibrium between habitat suitability and the occurrence of species, we can expect that species track temperature change equally[33,35], hence tropicalization would be similar to deborealization. However, in species with slow demography and limited dispersal, lags between climate change and distribution shifts can result in 'extinction debts'[36], where populations temporarily persist under unsuitable conditions, and 'colonization credits', where suitable locations are not occupied[32]. This latter pattern has been observed for instance in some studies on trees[32] and demersal fish[33], whilst other studies found equally responsive

shifts at both range boundaries in marine ectotherms[35], and prevalence of the tropicalization in demersal fish[15]. The prevalence in tropicalization vs deborealization might depend on biological traits of the community (size, dispersal capacity), phenotypic plasticity enabling populations to remain in sub-optimal habitats[37], or due to and seascape aspects (e.g., marine geomorphological features) limiting species dispersal and ocean connectivity. Thus, it can be expected that seas with limited connectivity to open oceans would constrain species abundance increases (tropicalization + borealization) over species abundance decreases (detropicalization + deborealization) processes.

The large geomorphological and ecological differences lead to large differences in the physical, biogeochemical and ecosystem responses of European Seas to climate change[38]. All European Seas are warming at least since the 1980's, but faster in the semi-enclosed basins of the Mediterranean Sea[39], Baltic Sea[40], and Black Sea (1982–2018)[41], compared to the NE Atlantic[39,42]. Here, we analysed the extent to which long-term trends in marine communities were related to ocean warming across biological groups and European regional seas. To do so, we used the CTI, which tracks the mean thermal affinity of a community[29], to quantify temporal rates of turnover of the relative abundance of species. We applied CTI to long-term time series from 65 biodiversity monitoring programmes spanning the last four decades (in the longest case), which accounts for 1817 species, including zooplankton, benthos, demersal and pelagic assemblages. Across the entire range of organisms and habitats, results show an average rate of increase in CTI of 0.23 °C decade⁻¹, meaning a consistent response of the marine communities to ocean warming in European seas. Furthermore, we explored the main underlying ecological processes driving temporal variation in CTI including: tropicalization, detropicalization, borealization, and deborealization[15]. We hypothesised that an increase in CTI with time linked to ocean warming may vary according to the biological groups, degree of ocean connectivity, and habitat type. Specifically, semi-enclosed sea basins with lower ocean connectivity to warmer waters are expected to experience less tropicalization compared to deborealization than is the case in the well-connected northeast Atlantic region. Our findings indicate that tropicalization dominated in Atlantic sites compared to semi-enclosed basins, supporting the expectations. Cross-region and cross-taxa comparisons may identify climate refugia as important reservoirs of biodiversity[43], hotspots of high biodiversity climate velocity[44] and sentinel systems[45].

## Results
We compared the rate of temporal change in CTI (CTI$_r$) and its underlying ecological processes in biodiversity time series collected over the last four decades among six biological groups (hard-bottom and soft-bottom coastal benthic communities, zooplankton, demersal crustaceans, cephalopods, and fish). Biodiversity time series included three European Seas (NE Atlantic, Mediterranean Sea, and Baltic Sea), three habitats (marine benthic or demersal, marine pelagic, and estuarine), and two basin types based on the presence or absence of dispersal barriers to ocean connectivity (non-enclosed sea, i.e., Atlantic Ocean, and semi-enclosed seas).

In almost all biodiversity sampling sites, sea surface temperature increased from 1980 to 2021 (Fig. 2a). On average, the warming rate of sea surface temperature (SST) was 0.32 °C decade⁻¹ ($t = 28.24$, $n = 1730$, $p < 0.0001$) (Fig. 2c) and the change of sea temperature from surface down to 100 m (ST$_{100m}$) was 0.15 °C decade⁻¹ ($t = 9.70$, $n = 1720$, $p < 0.0001$). CTI$_r$ trends mirrored those in ocean warming with an increase over time at most sites (80.0% of sites were positive, and 47.7% were significantly positive) (Fig. 2b; Supplementary Fig. 1, Supplementary Data 1). On average, the CTI also increased with time (Fig. 2c) at a rate of 0.23 °C decade⁻¹ ($t = 14.69$, $n = 1730$, $p < 0.0001$, using a linear mixed model) (Fig. 2c). The relation between CTI$_r$ and sea temperature change across sites was positive and significant on

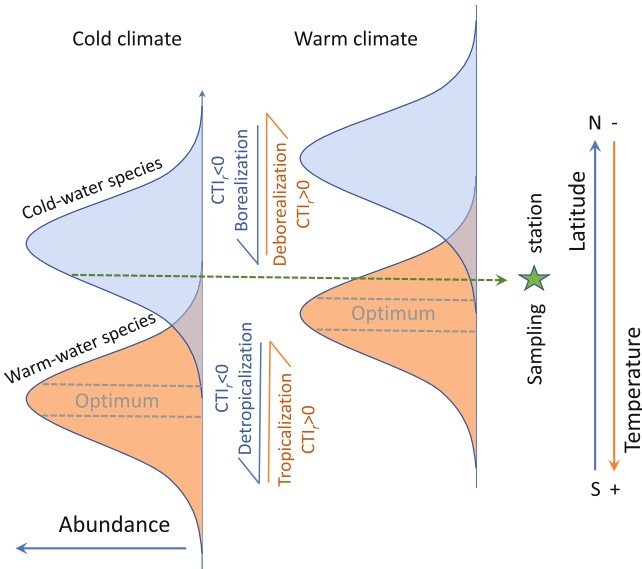

**Fig. 1 | Conceptualisation of species poleward distribution shift under warmer conditions.** Conceptualisation of poleward distribution shift and the expected abundance response curve of a cold- and warm-water species from cold to warm climate conditions, under the assumption of niche tracking. The sampling station illustrates how a long-term monitoring programme based on a permanent station is expected to detect changes in the abundance of species in a community affected by warming as a result of species' distribution shifts. At the community level, the processes of latitudinal shift triggered by warming at the sampling station can cause a positive rate of change in the Community Temperature Index (CTI$_r$) through the increase of warm-affinity species (tropicalization) and/or decrease of cold-affinity species (deborealization). Modified from Villarino et al.[79].

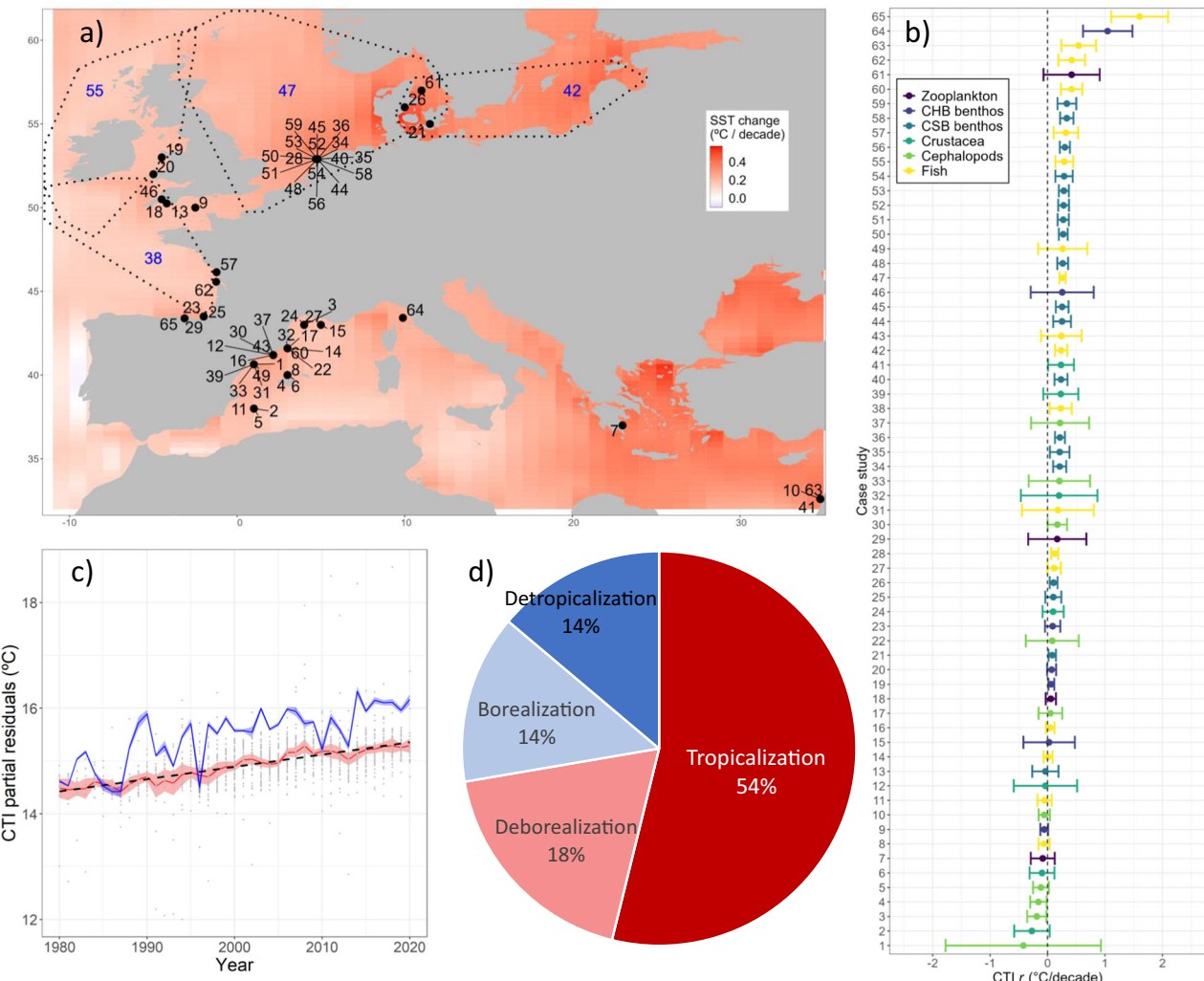

**Fig. 2 | Sea temperature and Community Temperature Index (CTI) trends in European seas. a** Mean Sea Surface Temperature (SST) trends from 1980 to 2020 in European seas; from GODAS (Global Ocean Data Assimilation System) data. Sampling site locations are shown in black circles (labelled in black) and polygons (labelled in blue); for sites labelling see Supplementary Data 1. Map source: geom-sf function from ggplot2 R package[97]. **b** Mean rate of change in Community Temperature Index (CTI) trends over time for each biodiversity time series with corresponding 95% confidence intervals. Sample size of the confidence intervals are defined by the number of years sampled (See Supplementary Data 1). For site labelling, see Supplementary Data 1. Source data are provided as a Source Data file. **c** Partial residuals of CTI across time, calculated as CTI minus the random effect of sampling site of the mixed model with year as fixed effect and site as random effect, see "Methods" section. In blue, partial residuals of SST across time. Grey points: partial residuals. Black dotted line: fixed effect of the linear mixed model. In red and shaded pink: annual mean and confidence interval of CTI partial residuals. In blue: annual mean and confidence interval of SST partial residuals. **d** Percentage of the prevailing underlying process ((de)tropicalization, and (de)borealization) over all biodiversity time series. CHB Coastal Hard Bottom, CSB Coastal Soft Bottom.

average for both SST ($t = 7.47$, $n = 1730$, $p < 0.0001$) and $ST_{100m}$ ($t = 4.33$, $n = 1720$, $p < 0.0001$), indicating that changes in marine communities can be associated with ocean warming.

We compared $CTI_r$ of marine communities across several factors (biological group, habitat, region, and basin type) (Supplementary Fig. 2). By biological group, $CTI_r$ trends were significantly positive for coastal soft-bottom benthos, fish, and zooplankton communities, with no significant change for demersal crustaceans and cephalopods (Fig. 3a) (Table 1). By habitat type, CTI trends were significantly positive in all habitat types (Fig. 3b and Table 1). Estuaries showed the highest $CTI_r$ value. $CTI_r$ was significantly positive in the NE Atlantic and Baltic, but not significant in the Mediterranean Sea (Fig. 3c and Table 1). The mean $CTI_r$ was also significant in both categories of basins (connectivity), with slightly higher values in non-enclosed seas compared to semi-enclosed seas (including estuarine types) (Fig. 3d and Table 1). The selection of factors identified the region and habitat as the best model (AICc = 1756.8). Diagnostic plots for the residuals of the selected

factors were checked, indicating model reliability (Supplementary Fig. 3).

The underlying ecological processes explaining the increase in CTI can generally be attributed to the prevalence of tropicalization or deborealization in most of sites (76.9% of sites), whilst detropicalization and borealization dominated at fewer sites (23.1%) (Fig. 2d). On a per-species basis (encompassing 5324 cases of 1817 species), the tropicalization and deborealization account for 59.3% of the overall process intensity (Fig. 4). Among the four processes considered, tropicalization was the most frequent (53.8%) on a per-site basis, followed by deborealization (18.5%) (Fig. 2d). In those biodiversity time series where the $CTI_r$ increased (i.e., 52 sites, 80% of sites), we analysed tropicalization relative to deborealization across biological groups, habitat, regions, and basin types (Table 2). By biological group, tropicalization relative to deborealization was most frequent for coastal benthos (soft- and hard-bottom), whilst the other groups showed more equal process intensities (Table 2). By habitat type,

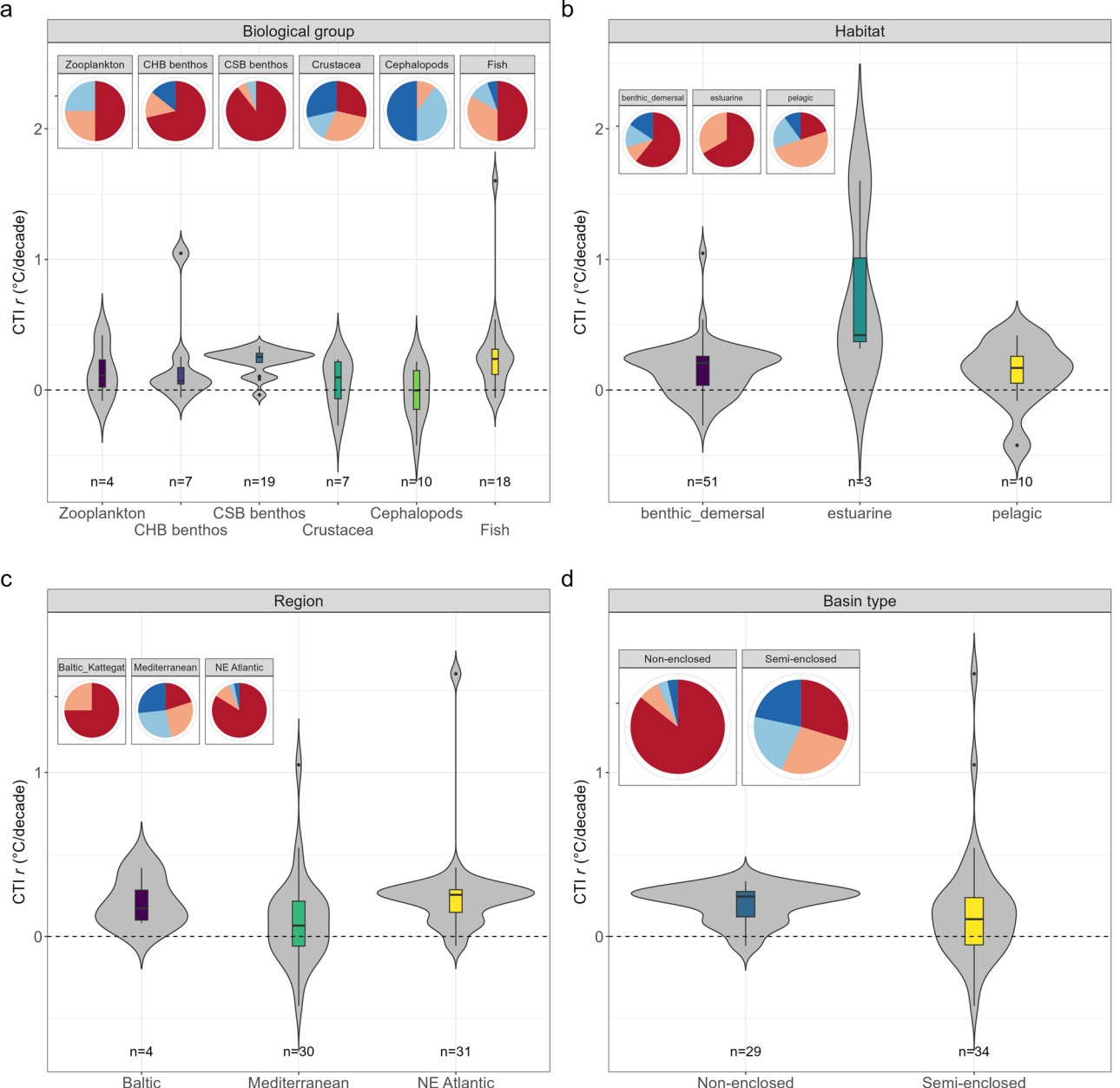

**Fig. 3 | Change in CTI across biological group, habitat, sea region, and basin type.** Boxplot and violin plots showing the change in the Community Temperature Index (CTIr) by biological group (**a**), habitat (**b**), sea region (**c**), and basin type (**d**), as well as the CTI underlying process dominance for each factor (pie charts). The bottom and top of the boxplots are the lower (Q1, i.e. 25%) and upper (Q3, i.e. 75%) quartiles, and the band inside the box is the median. The whiskers extend up to 1.5-fold the interquartile range (Q3−Q1) from the box. The violin plot shows the kernel probability density of the data at different CTIr values. CHB coastal hard bottom, CSB coastal soft bottom. Pie charts were computed estimating the percentage of the prevailing underlying process ((de)tropicalization, and (de)borealization) in each case study and by factor (Biological group, habitat, region, basin type). Pie chart legend: tropicalization (red), deborealization (pink), detropicalization (light blue), borealization (dark blue). The sample size (*n*) indicates the number of sites to the corresponding level of each factor. Source data are provided as a Source Data file.

tropicalization only dominated in benthic/demersal habitats (Table 2). By region, tropicalization dominated in NE Atlantic, whilst Baltic Sea and specially the Mediterranean presented more equal intensity (Table 2).

By basin type (i.e., ocean connectivity), tropicalization relative to deborealization strongly dominated the community responses in the non-closed seas (i.e., Atlantic Ocean), whilst semi-enclosed seas presented more equal process intensities (Table 2 and Fig. 5a). Comparing factors individually, basin type explained the greatest amount of variation of the process intensity according to the lowest AICc. Two models were selected as the best ones with the same AICc (i.e., 33.9):

(1) habitat and region, and (2) habitat, region, and basin type. The results of the analysis of process intensity at per-species basis were qualitatively similar to those at per-site basis (Supplementary Table 1).

Species' abundance increases (i.e., tropicalization + borealization) relative to species' abundance decreases (i.e., detropicalization + deborealization) dominated in non-closed compared to semi-enclosed seas (*p* = 0.00018; Fig. 5b).

## Discussion

Using a large number of long-term marine biodiversity time series from European seas, we quantitatively compare the underlying

**Table 1 | Community Temperature Index temporal change (CTI$_r$) by factors and its levels**

| Factor | Level | $n$ | CTI, mean (°C y$^{-1}$) | CTI, SE | DF | $t$-value | $p$-value | AICc |
|---|---|---|---|---|---|---|---|---|
| Biological group | CHB benthos | 7 | 0.01171 | 0.00991 | 1659 | 1181 | 0.2378 | 1841.87 |
| | CSB benthos | 19 | 0.02553 | 0.00188 | 1659 | 13,546 | **<0.0001** | |
| | Zooplankton | 4 | 0.01641 | 0.00629 | 1659 | 2605 | **0.0093** | |
| | Demersal crustaceans | 7 | 0.00464 | 0.00590 | 1659 | 0786 | 0.4315 | |
| | Cephalopods | 10 | −0.00792 | 0.00536 | 1659 | −1476 | 0.1400 | |
| | Fish | 18 | 0.03234 | 0.00299 | 1659 | 10,831 | **<0.0001** | |
| Habitat | Benthic / demersal | 52 | 0.02061 | 0.00149 | 1661 | 13,820 | **<0.0001** | 1843.39 |
| | Estuarine | 3 | 0.07732 | 0.00624 | 1661 | 12,387 | **<0.0001** | |
| | Pelagic | 10 | 0.01822 | 0.00487 | 1661 | 3741 | **0.0002** | |
| Region | Baltic Sea | 4 | 0.02037 | 0.00526 | 1662 | 3872 | **0.0001** | 1744.97 |
| | Mediterranean Sea | 30 | 0.00512 | 0.00306 | 1662 | 1674 | 0.0943 | |
| | Northeast Atlantic | 31 | 0.02854 | 0.00171 | 1662 | 16,668 | **<0.0001** | |
| Basin type | Non-enclosed | 28 | 0.02482 | 0.00189 | 1663 | 13,069 | **<0.0001** | 1850.98 |
| | Semi-enclosed seas | 37 | 0.01876 | 0.00261 | 1663 | 7178 | **<0.0001** | |

Test for significant differences in mean CTI$_r$ from zero, according to linear mixed models. t- and p-values correspond to two-sided Wald test. Significant p-values (p < 0.05) are in bold.
*HB* hard bottom, *SB* soft bottom, *SE* standard error, *AICc* Akaike's Information Criterion corrected, *DF* degrees of freedom.

ecological responses of marine communities to ocean warming across groups of marine organisms, habitats, and ocean connectivity. We detected community changes in relation to ocean warming in all European seas analysed over the last four decades (1980 to 2022). In nearly all biodiversity sampling sites, the sea surface had warmed at an average rate of 0.32 °C decade$^{-1}$ and 0.15 °C decade$^{-1}$ in the integrated upper 100 m, faster in the Mediterranean Sea and Baltic Sea than in the NE Atlantic Ocean, in agreement with previous studies[39]. Between 1982 and 2010, for instance, coastal SST warmed at 0.2–0.3 °C decade$^{-1}$ off the southwest European coast, and up to 0.3–0.7 °C decade$^{-1}$ in the Norwegian and North Seas[46,47]. In the Bay of Biscay, warming trends between 0.10 to 0.25 °C decade$^{-1}$ started in the 1980s[42,48], with a greater increase at the surface and a deepening of the 14 °C isotherm[49]. Sea warming in European semi-enclosed seas was stronger relative to open ocean: 0.4 °C decade$^{-1}$ in the Baltic Sea (1978–2007)[40], 0.9 °C decade$^{-1}$ in the Mediterranean[39], and 0.6 °C decade$^{-1}$ in the Black Sea (1982-2018)[41].

Across the entire range of organisms and habitats, the average rate of increase in CTI was 0.23 °C decade$^{-1}$, meaning a consistent response of the marine communities to ocean warming in European seas. The difference between CTI and sea temperature changing rates may represent a slight temporal lag in the response of communities to ocean warming, as reported in terrestrial groups[29], since population-level acclimatisation may require several years for species with lifespans longer than one year (e.g., fishes, many crustaceans, cephalopods, and other macroinvertebrates). The active selection of specific ranges in water temperatures is particularly critical during fish reproduction[50], but affects the entire life cycle[51]. The correlation between the rates of change in CTI and concomitant warming in both surface waters and the water column down to 100 m is evidence of broadscale impacts of climate change throughout European regional seas. However, the relationship between CTI$_r$ and temperature change may not necessarily be linear, since species' thermal performance curves are asymmetrical (typically dome-shaped), and are often life-stage specific. Furthermore, other than long-term warming, species may also respond to marine heatwaves[52,53] and extreme weather events[45], or be affected indirectly by biotic interactions[54–56]. In addition, the community acclimatisation pathway to warming analysed here is primarily attributed to latitudinal shifts (active or passive), although other processes such as vertical migrations (or persistent changes in depth) in highly mobile species[19,57], or phenological adjustments[11,58] may also account for the observed patterns. Phenology changes allow timing adjustments of seasonal life cycle events such as

spawning of mainly sessile species or those with geographical attachment, which may counterbalance latitudinal shifts[59].

Among the underlying processes contributing to the observed changes in CTI, tropicalization and deborealization dominated over detropicalization and borealization. Moreover, the intensity of tropicalization was more than three times higher than that of deborealization at per-site basis. However, the relative importance of each process underpinning CTI changes over time varied across habitats, regions, and ocean connectivity. In those sites where CTI increased over time, tropicalization prevailed (with respect to deborealization) in non-enclosed waters, NE Atlantic, and in benthic and demersal habitats. Tropicalization can occur in well-connected seas where species can disperse from more tropical areas to colonize new habitats at higher latitudes[60], whilst in semi-enclosed basins, colonization is partially limited by landmasses. This may explain why tropicalization prevailed in over half of the sites, particularly those in the Atlantic Ocean, whilst both deborealization and tropicalization occurred in semi-enclosed seas. Furthermore, the relative importance of tropicalization and deborealization in European seas seems to depend on constraints on species' movement and dispersal across the seas (i.e., ocean connectivity), rather than on their coarse biological groups. On the Atlantic coast, warm-affinity (tropical) species can move from the (sub)tropics poleward as the water warms, but this natural source of warmer-affinity species is partially limited by landmasses (or might require longer time spans) in the Baltic or the Mediterranean Sea.

Within the biological groups considered, species can have a very different dispersal capacity according to their propagule-dispersing strategy, pelagic larval durations[61] and adult active mobility, especially in macroinvertebrates[25,26,62], which may explain why biological group was not a relevant factor in both CTI$_r$ and underlying ecological processes. In the fish community, which is one of the most well-represented groups in our data set, an ad hoc analysis revealed that tropicalization is dominant in the NE Atlantic (55%) compared to semi-enclosed seas (32%). This is probably because the capacity of fish individuals to reach semi-enclosed seas and to colonize new habitats is limited by geographical constraints and dispersal barriers, which is also supported by the results obtained from species abundance change in relation to their thermal affinities for all biological groups (Fig. 5b). Although our analysis is limited by replication of the ocean connectivity that only encompasses two semi-enclosed basin seas, three estuaries, and one non-enclosed system (i.e., NE Atlantic), we provide examples and previous studies that support this explanation for the observed patterns. Relevant examples of decreases in cold-

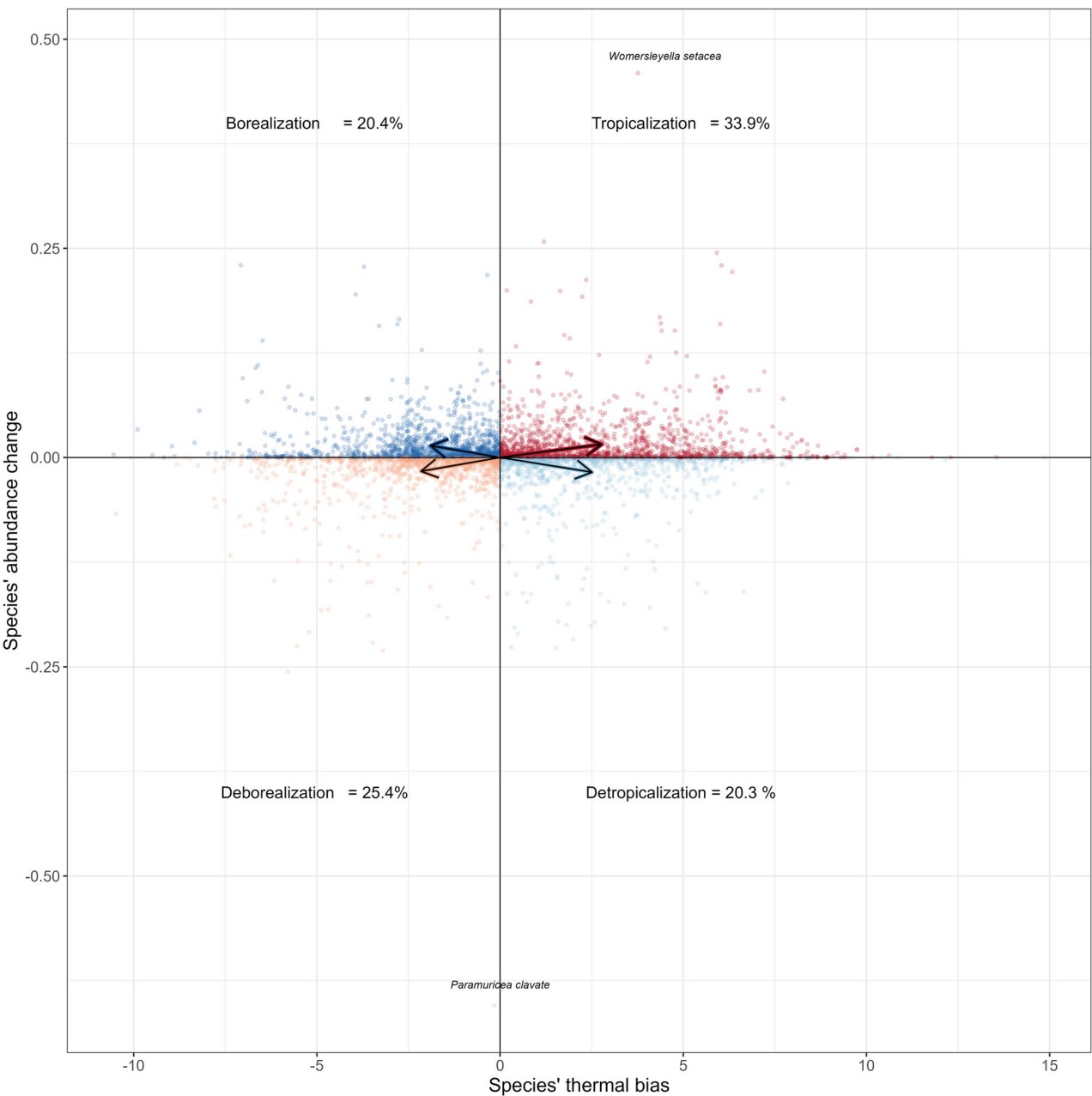

**Fig. 4 | Species' abundance change against species' thermal bias (species thermal preference - CTI) for all case studies sites.** Arrows represent the mean value of Species' thermal bias with respect to Species' abundance change for each underlying process (i.e. tropicalization, borealization, detropicalization, deborealization). Extreme values in Species' abundance change are represented by tropicalization in the tropical red alga *Womersleyella setacea* and deborealization of the temperate-water species of gorgonian *Paramuricea clavate* (see species' names labels), both in the coastal hard-bottom coralligenous communities in the Western Mediterranean.

affinity species in the Mediterranean and Baltic Seas are the ecologically and commercially important stocks of European sardine (*Sardina pilchardus*)[63] and Atlantic cod (*Gadus morhua*)[64]. Emigration or mortality of these cold-affinity species driven by warming could result in population crashes or local functional extinctions with important socio-economic consequences for fisheries[65]. Similarly, in Gironde estuarine fish communities, high deborealization may be due to the decline in abundance of cold-affinity species, corresponding mainly to diadromous fishes (*Platichthys flesus, Alosa fallax, Anguilla anguilla, Osmerus eperlanus*), in line with the decline in abundance of cold-water fish species in other European estuaries[66]. Due to their complex life cycles and homing behaviour[67], experiencing the effects of changes in climate conditions in both marine and freshwater habitats and across

different life stages, diadromous species are especially sensitive and vulnerable to climate change[68], probably in line with high $CTI_r$ found in estuaries.

The Mediterranean Sea was the region with higher deborealization relative to tropicalization. An example, supported by recent studies[63,64], was the intertidal hard-bottom community in the Ligurian Sea (northwest Mediterranean), dominated by the fucoid *Ericaria amentacea* that provides habitat to a variety of macroalgae and sessile invertebrates[69,70]. One exception where both tropicalization and deborealization are very strong in the semi-enclosed is the Levant basin, at the southeastern corner of the Mediterranean, where there is a conduit for tropical species from the Indo-Pacific via the Suez Canal[23,71]. Indeed, in this area, thermophilic species increasingly prevail

**Table 2 | Underlying process for positive temporal rate of Community Temperature Index (CTIᵣ>0)**

| Factor | Level | n | Tropicalization minus Deborealization mean (°C y⁻¹) | Tropicalization minus Deborealization SE | DF | t-value | p-value | AICc |
|---|---|---|---|---|---|---|---|---|
| Biological group | CHB benthos | 6 | 0.3020 | 0.1489 | 46 | 2.028 | **0.0484** | 49.17 |
| | CSB benthos | 18 | 0.5422 | 0.0859 | 46 | 6.306 | **<0.0001** | |
| | Zooplankton | 3 | 0.1716 | 0.2106 | 46 | 0.815 | 0.4193 | |
| | Demersal crustaceans | 4 | −0.1107 | 0.1841 | 46 | −0.607 | 0.5468 | |
| | Cephalopods | 5 | −0.0647 | 0.1647 | 46 | −0.397 | 0.6935 | |
| | Fish | 16 | 0.0537 | 0.0912 | 46 | 0.589 | 0.5590 | |
| Habitat | Benthic / demersal | 40 | 0.3413 | 0.0613 | 48 | 5.566 | **<0.0001** | 53.03 |
| | Estuarine | 3 | 0.0042 | 0.2239 | 48 | 0.019 | 0.9850 | |
| | Pelagic | 8 | −0.1429 | 0.1371 | 48 | −1.042 | 0.3030 | |
| Region | Baltic Sea | 4 | 0.2903 | 0.1809 | 49 | 1.605 | 0.1150 | 41.85 |
| | Mediterranean Sea | 19 | −0.0731 | 0.0832 | 49 | −0.881 | 0.3830 | |
| | Northeast Atlantic | 29 | 0.4278 | 0.0672 | 49 | 6.366 | **<0.0001** | |
| Basin type | Non-enclosed seas | 26 | 0.4767 | 0.0693 | 50 | 6.873 | **<0.0001** | 37.63 |
| | Semi-enclosed seas | 26 | −0.0083 | 0.0693 | 50 | −0.119 | 0.9050 | |

Tropicalization minus deborealization analysed with linear models by factors. t- and p-values correspond to two-sided Wald test. Significant p-values (p < 0.05) are in bold.
*CHB* coastal hard bottom, *CSB* coastal soft bottom, *SE* standard error, *AICc* Akaike's Information Criterion corrected, *DF* degrees of freedom.

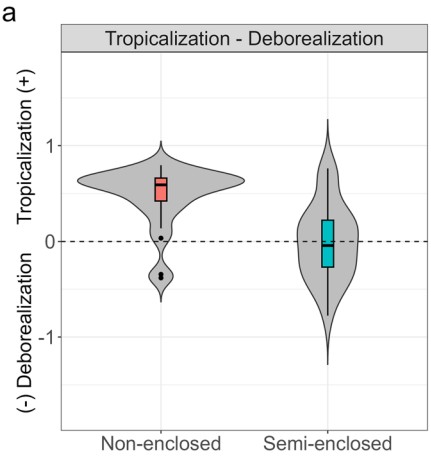

**Fig. 5 | Tropicalization minus deborealization by basin type.** Boxplot and violin plots showing (**a**) Community Temperature Index (CTI) underlying processes of tropicalization minus deborealization for positive CTIr by basin type, (**b**) (Tropicalization + Borealization) minus (Detropicalization + Deborealization) by basin type for all sites. The bottom and top of the boxplots are the lower (Q1) and upper (Q3) quartiles, and the band inside the box is the median. The whiskers extend up to 1.5- fold the interquartile range (Q3−Q1) from the box. The violin plot shows the kernel probability density of the data at different process intensity values. Trop Tropicalization, Bor Borealization, Det Detropicalization, Deb Deborealization. Sample size 28 (non-enclosed), 37 (semi-enclosed). Source data are provided as a Source Data file.

in many groups while deborealization also occurs as many native species either decrease in abundance or completely disappear from local communities[23,71–73]. There is also recent experimental evidence that key native species collapsed by a combination of warming and competition with tropical alien invaders, and that aliens are more resilient to warming than native species in the region[74]. The significant ocean warming in the eastern Mediterranean in 1990-2020 coincided with a significant increase in the CTI over time of the community of 145 demersal fishes, tracking their thermal niches over time at a rate of 0.54 °C decade⁻¹. Thus, the deborealization trend found here may be a product of competition between native and non-indigenous species with similar traits or niches, and/or the warming-induced reductions in the fitness of thermally-sensitive native species. A recent study suggests that the latter may be more important than the former[71].

One of the most diverse communities in the Mediterranean Sea analysed here was the coralligenous assemblages[75]. The analysis of CTI

of coralligenous assemblages in the Western Mediterranean over the 2008–2020 period characterised by significant sea warming (0.21 °C decade⁻¹) indicated that tropicalization was more important than other processes in terms of changes in relative abundance. The tropical red alga *Womersleyella setacea* present in this assemblage was the species with highest tropicalization in the overall dataset (Fig. 4); this species is causing increasing concern in the Mediterranean Sea because of its invasive behaviour[76]. Furthermore, a temperate-water species of gorgonian (*Paramuricea clavata*) has also dramatically decreased (60%), which corresponds to the most deborealized case detected here (Fig. 4). Previous studies in the Western Mediterranean Sea have demonstrated that marine heatwaves can cause long-term alterations to the functional trait composition of coralligenous assemblages, such as the habitat-forming octocorals[77,78]. Gorgonians, which are some of the most important habitat-formers in coralligenous assemblages, have a general low capacity to recover from marine heatwaves, even after more than a decade[78].

Although few long-term time series of zooplankton communities have been analysed here to make definitive conclusions, CTI changed less consistently with sea warming compared to changes observed for fish, in agreement with the analysis of Burrows et al. [19]. Tropicalization was the main underlying ecological process explaining the trends in zooplankton communities, suggesting thermal niche tracking in response to rates of increase in annual temperature. Previous studies on a subset of these data reported high turnover in the relative abundance of species in the copepod community, with species similarity decreasing over time associated with both niche descriptors (including temperature) and demographic stochastic processes[79].

In summary, a significant portion of marine communities and sites examined across European seas showed a clear response to ongoing ocean warming which, in most cases, favoured mainly warm-water species (tropicalization) in combination with decreases in cold-water species (deborealization). In the Northeast Atlantic, tropicalization prevails probably because the seascape is open with a pool of warm-affinity species that can arrive from lower latitudes, whilst land barriers in the Baltic and the Mediterranean Seas appears to partially limit such colonization, except in the southeast Mediterranean that is connected to the Indo-Pacific Ocean through the Suez Canal. The relative importance between tropicalization and deborealization seems to depend mainly on species habitat and on the ocean connectivity of the sea, although life-history traits such as body size, lifespan, and thermal tolerance have also been previously identified as relevant at the species level[15,16,19]. Further analyses at the global scale that include distinct basins with a wider degree of ocean connectivity are needed to confirm our conclusions. Cross-region and cross-taxa comparisons enabled the identification of the semi-enclosed Mediterranean and Baltic Seas as the most vulnerable European marine ecosystems facing climate change, since they are those with highest water warming rates and where communities are experiencing local loss of biodiversity through deborealization processes.

## Methods

### Temperature data

To provide a common long-term dataset of sea temperatures through water column (surface and down to 100 m), data from the NCEP Global Ocean Data Assimilation System (GODAS) (www.cpc.ncep.noaa.gov/products/GODAS/) were extracted for each site for the 1980−2020 period. GODAS provides monthly water temperature data to a 0.333° × 1° latitude-longitude grid. Annual data from the grid centred to each biodiversity sampling site was extracted, averaging a spatial window according to the area of the site. We analysed the slope of the temperature trends and its significance using a linear mixed model[80] of sea temperature with year as fixed effect, the sampling sites as random effect, and adding a temporal autoregressive function. $t$- and $p$-values of the two-sided Wald test for the estimated slope coefficients are provided.

### Biodiversity time series

We used 65 long-term time series corresponding to 1817 species taxonomically identified at species level (71 zooplankton species[79,81], 238 coastal hard-bottom benthic species[18,77,82,83], 923 coastal soft-bottom benthic species[83–85], 63 cephalopod species[86,87], 104 demersal crustacean species[86,87], 418 fish species[86–92]). Fish includes pelagic, demersal, and estuarine fish species. Demersal crustaceans were collected offshore from bottom trawl fishing and surveys. Cephalopods includes demersal and pelagic species collected offshore from bottom trawl and purse-seine fishing and surveys. Coastal soft-bottom benthic macroinvertebrates includes species sampled in coastal intertidal or subtidal soft-bottom sites, using grab or box corer samples. Coastal hard-bottom benthos includes macroinvertebrates, macroalgae, lichens, and coralligenous assemblages in intertidal or subtidal, coastal, hard-bottom sites. Zooplankton includes mainly copepods

and other species collected with zooplankton net hauls. Time series ranges over varying durations during the last four decades (from 1980 to 2022 in the longest case). Most of time series span more than 15 years (92%), with median of 25 years, minimum of 9 years, and 20 and 39 years the 25% and 75% quartiles, respectively. Detailed information on each time series such as data sources, sampling procedures, time series length, biological community, and location is provided in Supplementary Information 1.

### Species relative abundance turnover: Community Temperature Index through time

To test whether temporal changes in the composition of species in a community respond to warming according to their thermal preferences, we used the CTI, which is a measure of the average thermal affinity of ecological communities, weighted by the relative abundance[29,30]:

$$CTI = \sum_{s=1}^{n} T_s \times \log(A_s + 1) \qquad (1)$$

where $n$ is the number of species in the community, $T_s$ is the temperature preference of each species ($s$) and $A_s$ is the relative abundance of species $s$ (i.e., the abundance of species $s$ divided by the total number of individuals in the community at a site). The thermal preferences were determined for each species by matching occurrence records collected from OBIS (Ocean Biogeographic Information System; www.iobis.org) with annual means of GODAS local SST during the 1980 − 2021 period[67]. OBIS records were quality-checked removing duplicates. To characterise the thermal preferences of the species local temperatures derived from GODAS were used, which was available from OBIS for each observation. The midpoint between the 5th and 95th percentile of the temperature distribution occupied by each species was calculated as a measure of central tendency of their realised thermal distribution[30]. Subsequently, CTI was calculated at each station using the thermal midpoint values for each species recorded weighted by their $log$(abundance + 1)[30]. CTI computation have been coded in R language[93] and available for one of the time series in a public repository (https://doi.org/10.5281/zenodo.10708267, https://doi.org/10.5281/zenodo.10708267).

Inter-annual temporal change rates in CTI ($CTI_r$) were estimated using two approaches: (1) for each site independently, and (2) jointly for all sites to provide an overall estimate of CTI change. For the first approach, we estimated the $CTI_r$ for each site. To avoid potential biases in the estimation of $CTI_r$ due to temporal autocorrelation[94], partial autocorrelation has been checked for each yearly data time series. Subsequently, $CTI_r$ was estimated fitting linear models using generalised least squares, and adding an autoregressive function when autocorrelation was detected. For the second approach, we tested whether the CTI change through time for all sites on average is different from zero using a linear mixed model of CTI with year as the fixed effect, the sampling sites as the random effect, and adding a temporal autoregressive function. The relation between CTI and sea temperature (surface and 100 m integrated column) through time and across sites were tested using linear mixed models with site as random effect, and adding a temporal autoregressive function. t- and p-values of the two-sided Wald test for the estimated slope coefficients are provided.

### Tropicalization and borealization of communities

CTI changes were decomposed into four underlying process following McLean et al. [15]: tropicalization (increasing warm-affinity species), deborealization (decreasing cold-affinity species), borealization (increasing cold-affinity species), and detropicalization (decreasing warm-affinity species). This categorisation is computed at per-species basis by (i) calculating species' thermal bias (thermal preference – $CTI_{station}$) and species' abundance change, and (ii) assign the corresponding process to each species, e.g., tropicalization if both species'

thermal bias and abundance change are positive, deborealization if both species' thermal bias and abundance change are negative. The intensity of the ecological processes underlying the temporal change in CTI at each station and overall datasets were also examined by (i) calculating the difference between each species' thermal preference and the mean of the community, (ii) multiplying this value by each species' change in abundance, and (iii) taking the sum of the resulting values for all species within each process[15]. Thus, underlying process at per-site basis refers to the community, e.g. tropicalization intensity of the community corresponds to an increase in the number of warm-affinity species, and deborealization to a decrease in cold-affinity species. According to the latitudinal position of the sampling, a given species can be a cold-affinity species at a certain latitude and a warm-affinity species at another latitude. Increases in CTI are expected to occur when the combination of tropicalization and deborealization is stronger than the combination of borealization and de-tropicalization. CTI analysis and underlying process (i.e., tropicalization and borealization) have been coded in R language[93] and available in a public repository (https://doi.org/10.5281/zenodo.10708267, https://doi.org/10.5281/zenodo.10708267). As an overall statistic, the percentages of the prevailing underlying process ((de)tropicalization, and (de)borealization) over all biodiversity time series were also calculated.

### Analysis across regions and biotic groups

We compared the $CTI_r$ of marine communities across different factors (biological groups, habitat, sea, and basin type) and their levels (Biological groups: coastal hard-bottom benthos, coastal soft-bottom benthos, zooplankton, demersal crustaceans, cephalopods, fish; Habitat: benthic / demersal, estuarine, and pelagic; Sea: Baltic (including Kattegat), Mediterranean, Northeast Atlantic, and Basin type: Non-enclosed (Atlantic Ocean), Semi-enclosed (Mediterranean, Baltic Sea, and estuarine types). The comparison was based on linear mixed models of CTI with the interaction of year and factor as the fixed effect, the sampling sites as the random effect, and adding a temporal autoregressive function to test if the $CTI_r$ mean of a given level differs from zero. t- and p-values of the two-sided Wald test for the estimated coefficients are provided. To identify the most representative factors, we selected the best model using the Akaike's Information Criterion corrected (AICc) by comparing all combinations[95]. Diagnostic plots for the residuals of the selected factors were checked to ensure model reliability.

As positive CTI changes correspond to an increased prevalence in warm-affinity species (i.e., tropicalization), and/or a decrease in cold-affinity species (i.e., deborealization), the aim here is to test if the intensity of tropicalization minus deborealization at per-site basis varies across levels of the different factors. The reason why tropicalization may prevail (or not) over deborealization might depend on biological traits (size, dispersal capacity), and seascape aspects (e.g., marine geomorphological features) limiting species dispersal and ocean connectivity, which has been recently debated. For positive CTIr, we tested different factors. Therefore, we compared the means of intensity of underlying processes of increased $CTI_r$ (in particular, tropicalization *minus* borealization) of marine communities across different factors and their levels using linear models[96] and selected the best model using the AICc by comparing all combinations, or with forward stepwise selection in case of singularities caused from interaction among factors[95]). For comparison purposes, we also modelled the intensity of underlying processes at per-species basis with a linear mixed model for $CTI_r > 0$ as a function of factors, with sites as random effect.

It can be also expected that seas with limited connectivity to open oceans would constrain species abundance increases (i.e., tropicalization and borealization) over abundance decreases (i.e., detropicalization and deborealization) processes in both CTI increase or decrease. Therefore, for basin type, we also tested whether species abundance increase (tropicalization + borealization) *minus* abundance decrease (detropicalization + deborealization) processes varied across levels at per-site basis.

All these analysis have been coded in R language[93] and available in a public repository (https://doi.org/10.5281/zenodo.10708267, https://doi.org/10.5281/zenodo.10708267).

### Reporting summary

Further information on research design is available in the Nature Portfolio Reporting Summary linked to this article.

## Data availability

Biodiversity original data (i.e., species abundance at each year for each site survey) is subject to restrictions as it pertains to the corresponding institution. Certain original data is publicly available (DATRAS-ICES[88], Danish marine monitoring[84], soft-bottom benthos in Basque estuaries[98]) or from the corresponding author upon request. Data generated during the study and that support their findings (all CTI time series, i.e. CTI per year for each site, and all underlying process scores ((de)tropicalization, (de)borealization) at per-species and per-site basis) are available in a public repository[98] under accession code https://doi.org/10.5281/zenodo.10708267 (https://doi.org/10.5281/zenodo.10708267), and in Supplementary Data 1. Source data are provided with this paper.

## Code availability

All codes are available in a public repository[98] (https://doi.org/10.5281/zenodo.10708267, https://doi.org/10.5281/zenodo.10708267).

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

## Acknowledgements

This study has been supported by the European Union's Horizon 2020 research and innovation programme under grant agreement No 869300 (FutureMARES project) (G.C., E.V., J.A.F-S., M.A.P., M.Lin., C.D., D.G-G., G.R., L.B-C., C.R., M.C., F.R., G.R., P.B., M.Lep., J.L., A.B., N.Mi., J.G., F.B., A.M.Q., F.V., A.I. and I.U.), and by the Urban Klima 2050 – LIFE 18 IPC 000001 project, which has been received funding from European

Union's LIFE programme (G.C., E.V., L.I., A.B., A.U. and M.R.). Additional financial support was obtained from the Basque Government (PIBA2020-1-0028 & IT1723-22). We thank the NOAA Climate Prediction Center for providing Sea Temperature data through the NCEP Global Ocean Data Assimilation System (GODAS) www.cpc.ncep.noaa.gov/products/GODAS. We also thank Ocean Biodiversity Information System (OBIS) for providing global occurrences of the biological group studied here. Data from the Basque Country were obtained from the Basque Water Agency (URA) monitoring network, through a Convention with AZTI, and by the Bilbao Bizkaia Water Consortium. M.C., J.G., D.G.-G. and F.R. acknowledge the 'Severo Ochoa Centre of Excellence' accreditation (CEX2019-000928-S). Authors M.H. and M.Lin. are grateful for the support from ICES Working Group on Comparative Ecosystem-based Analyses of Atlantic and Mediterranean marine systems (WGCOMEDA) for this research. This paper is contribution n° 1208 from AZTI, Marine Research, Basque Research and Technology Alliance (BRTA).

## Author contributions

G.C., E.V., M.Lin., M.McL. and A.A. conceived and designed the research. G.C., E.V. and M.A.P. wrote the main text of the manuscript. G.C., E.V., L.I., C.D., D.G-G., P.d.l.B., F.R., G.R., E.Y., L.B-C. and C.R. analysed the data. G.C., E.V., M.Lin., M.McL, A.A., C.D., D.G-G., A.v.L., G.R., L.B-C., C.R., M.H., M.C., F.R., G.R., E.Y., P.d.l.B., M.A.P., M.Lep., C.D., J.L., A.B., N.Mi., J.G., F.B., J.C., A.M.Q., F.V., A.I., I.U. and J.A.F-S. interpreted and discussed the results. M.Lin. provided data on demersal fish in the North-East Atlantic. M.H. provided processed data on demersal and benthic fish, cephalopods, and crustaceans of the Iberian Mediterranean coast. M.C. and F.R. provided data on pelagic, demersal, and benthic fish, cephalopods, and crustaceans in the Western Mediterranean. G.R. and E.Y. provided data on demersal and benthic fish, cephalopods, and crustaceans in the Eastern Mediterranean. A.v.L and M.A.P. provided data on fish in Wadden Sea. P.d.l.B. and M.A.P. provided data on soft-bottom benthos in Wadden Sea. M.Lep., C.D., A.B'A. and J.L. provided data on fish in French estuaries. A.B., A.U. and M.R. provided data on fish in Basque estuaries. N.Mi. provided data on hard bottom benthic intertidal communities in UK coastline. J.G., D.G-G., C.L. and N.Ma. provided data on hard-bottom coralligenous communities. F.B., L.B-C. and C.R. provided data on hard-bottom benthos in the Ligurian Sea. A.B. and I.M. provided data on benthos in the Bay of Biscay. J.C. provided data on soft-bottom benthos in the Kattegat. A.J.McE. and A.M.Q. provided data on soft-bottom benthos (L4). J.C., F.V., A.I., I.U., S.Z., A.M.Q., A.J.McE. and P.J.S. provided data on zooplankton. All authors revised the manuscript.

## Competing interests

The authors declare no competing interests.

## Additional information

Guillem Chust [1] ✉, Ernesto Villarino[1,2], Matthew McLean [3], Nova Mieszkowska [4,5], Lisandro Benedetti-Cecchi[6], Fabio Bulleri [6], Chiara Ravaglioli[6], Angel Borja [1], Iñigo Muxika [1], José A. Fernandes-Salvador [1], Leire Ibaibarriaga[1], Ainhize Uriarte[1], Marta Revilla [1], Fernando Villate[7,8], Arantza Iriarte[8,9], Ibon Uriarte[8,9], Soultana Zervoudaki[10], Jacob Carstensen[11], Paul J. Somerfield [12,13,26], Ana M. Queirós [12,14], Andrea J. McEvoy [12], Arnaud Auber [15], Manuel Hidalgo[16], Marta Coll [17], Joaquim Garrabou[17], Daniel Gómez-Gras[18,19,20], Cristina Linares [19,20], Francisco Ramírez[17], Núria Margarit[19], Mario Lepage [21], Chloé Dambrine[21], Jérémy Lobry [21], Myron A. Peck [22], Paula de la Barra[22], Anieke van Leeuwen [22], Gil Rilov [23], Erez Yeruham[23], Anik Brind'Amour[24] & Martin Lindegren[25]

[1]AZTI Marine Research, Basque Research and Technology Alliance (BRTA), Txatxarramendi Ugartea z/g, 48395 Sukarrieta, Spain. [2]Oregon State University, College of Earth, Ocean and Atmospheric Science, Corvallis, USA. [3]Department of Biology and Marine Biology, Center for Marine Science, University of North Carolina Wilmington, Wilmington, NC, USA. [4]Marine Biological Association, Citadel hill, Plymouth, Devon PL1 2PB, UK. [5]University of Liverpool, Liverpool, UK. [6]Dipartimento di Biologia, Università di Pisa, CoNISMa, Via Derna 1, 56126 Pisa, Italy. [7]Department of Plant Biology and Ecology, Faculty of Science and Technology, University of the Basque Country (UPV/EHU), PO Box 644, E-48080 Bilbao, Spain. [8]Research Centre for Experimental Marine Biology and Biotechnology Plentzia Marine Station PiE-UPV/EHU, Areatza Pasalekua z/g, E-48620 Plentzia, Spain. [9]Department of Plant Biology and Ecology, Faculty of Pharmacy, University of the Basque Country (UPV/EHU), Paseo de la Universidad 7, E-01006 Gasteiz, Spain. [10]Institute of Oceanography, Hellenic Centre for Marine Research, Athens, Greece. [11]Aarhus University, Department of Ecoscience, Frederiksborgvej 399, DK-4000 Roskilde, Denmark. [12]Plymouth Marine Laboratory, Plymouth, UK. [13]University of Plymouth, Plymouth, UK. [14]University of Exeter, Exeter, UK. [15]IFREMER, Unité Halieutique Manche Mer du Nord, Laboratoire Ressources Halieutiques, 150 quai Gambetta, BP699, 62321 Boulogne-sur-Mer, France. [16]Spanish Institute of Oceanography (IEO, CSIC), Balearic Oceanographic Center (COB), Ecosystem Oceanography Group (GRECO), Moll de Ponent s/n, 07015 Palma, Spain. [17]Institute of Marine Science (ICM-CSIC),

Passeig Marítim de la Barceloneta, n° 37-49, 08003 Barcelona, Spain. [18]Hawai'i Institute of Marine Biology, University of Hawai'i at Mānoa, Kaneohe, Hawaii, USA. [19]Departament de Biologia Evolutiva, Ecologia i Ciències Ambientals, Universitat de Barcelona (UB), Barcelona, Spain. [20]Institut de Recerca de la Biodiversitat (IRBio), Universitat de Barcelona (UB), Barcelona, Spain. [21]INRAE, EABX Unit, Aquatic Ecosystems and Global Changes, 50 avenue de Verdun, 33612 Cestas, Cedex, France. [22]Department of Coastal Systems, Royal Netherlands Institute for Sea Research, PO Box 59, 1790 AB Den Burg (Texel), the Netherlands. [23]National Institute of Oceanography, Israel Oceanographic and Limnological Research (IOLR), Haifa, Israel. [24]Ecosystem Dynamics and Sustainability (UMR DECOD), IFREMER, Institut Agro, INRAE, Rue de l'Ile d'Yeu, Nantes, France. [25]Centre for Ocean Life, National Institute of Aquatic Resources, Technical University of Denmark, Kemitorvet, Building 202, 2800 Kgs Lyngby, Denmark. [26]Deceased: Paul J. Somerfield. ✉e-mail: gchust@azti.es

