## [Peer Review File · Nature Communications]

Cross-basin and cross-taxa patterns of marine community tropicalization and deborealization in warming European seasREVIEWER COMMENTS

Reviewer #1 (Remarks to the Author):

This is a neat paper that attempts to address an important ecological phenomenon. I particularly like the authors' intent to explore CTI change in the light its constituent components (topicalisation, demoralisation, etc.). As such, I would really like to see this work published. However, to properly assess the robustness of its findings, more detail about the analytical approaches is required—and depending on the outcome of that process, potentially also reanalysis of the data.

I have several major comments, as well as minor editorial suggestions.

Major comments:

1 – There were several unanswered questions about how the various groupings were constituted and what the group names really meant. Since this is a central focus of the manuscript, more detail/rationale is required here. Line-by-line examples follow.

*Line 183: Fig. 1b. Were crustaceans and molluscs included among the benthos, if not, why not (I don't see an answer in the Methods)?

*Lines 199–202: These results raise a further issue with separating molluscs and crustaceans from other benthos, unless the other benthos was sessile/burrowing, in which case, better names might be useful.

2 – There were several issues with the analyses. Although the authors demonstrate knowledge of most of the issues (e.g., temporal [and to some degree, spatial] autocorrelation, the need for mixed models, etc.), the application of techniques was inconsistent, and in some cases, (given the information provided in the manuscript) questionable. The most important of these is the apparent abandonment of the results from the mixed-effects model to produce test statistics in the main results table (Table 1), rendering their interpretation open to questions about the actual alpha-level (probability of committing a type I error) used in the analyses. There are several other issues listed line-by-line, below. Some of these issues might be resolved by more careful explanation of the rationale and philosophy of the individual analyses (models), but in other cases,

reconsidering the approaches to analyses might be warranted. Irrespective, it would be very useful to include diagnostic plots of the final fitted models in the Supplement, as well as a short section on caveats in the Discussion.

*Lines 203–205: Given that there is only one observation from estuaries, would it not be more rigorous to simply omit that datum than to point out that a single point had the strongest trend, albeit non-significant? I am surprised that the algorithm managed to estimate a p-value from a single point...although this would have depended on the structure of the model, I guess. Nevertheless, the inclusion of this point is somewhat problematic.

*Line 216–217: Fig. 3 – How were pies computed? On a per-species basis? How were these weighted when there were multiple locations/sites? This question further challenges the wisdom of including “estuary” as a habitat with only a single observation.

*Line 244–247: Table 2 and associated results – I’m not convinced that an ANOVA is the right way to go here. These are counts, so a log-linear analysis (saturated Poisson glm that includes interactions between grouping factors and process to test for independence of frequencies) seems more defensible...or perhaps a Poisson generalised mixed model that also includes the random effects?

*Lines 428–429: No consideration of temporal autocorrelation here? This needs to be addressed or at least listed as a caveat, with a brief mention of the likely consequences. You do this in Lines 465–467 for CTI_r, so why not temperature?

*Lines 433–439: A more detailed explanation for the soft-bottom/hard-bottom benthos vs mollusca/crustacea classifications are needed here, as is more detail on the durations of the time series (at least median, quartiles and range). Data are critical in this analysis, so such brief coverage is problematic.

*Lines 443–461: Move all of the technical information into a single paragraph before explaining how the metric is used. Otherwise, the reader is left wondering how Ts and A are determined.

*Lines 468–471: First, temporal autocorrelation seems to be missing from this analysis. Second, the description here clashes with information in the caption for Figure 2: if site/taxonomic group is the random factor, how can Fig. 2c be showing random effects of the mixed model? For models of biological group, surely the random effect should be site nested within group (i.e., the effect of site doesn’t disappear because you are more interested in taxa)? Finally, what is the effect of multiple tests, here? It is true that

corrections for multiple tests are usually deployed for t-tests and the like, but the principle holds for any sequence of hypothesis tests. Here, you are refitting different models to the same response data. This increases the risk of a Type I error. Given the marginal significance of several of the results in Table 1, this becomes more problematic. One solution might be to fit a model with all predictors included, then simplify on the basis of AIC or some other information-theoretic criterion, and interpret only the predictors that remain? Irrespective, this is a serious issue here. Proper diagnostic plots of the final fitted models would also be useful. Finally, since the results depend on these analyses, a clearer and more detailed explanation of the underlying models (including their underlying rationale/philosophy) would really help to clarify what is being done.

*Lines 475–484: This does not provide any real insight into how the frequency(?) pies were computed for presentation in Figures 2 & 3. Again, without understanding what these plots actually represent, it is impossible to properly assess their meaning.

*Lines 493–496: Here, it seems as if the mixed-effects models are simply ignored in favour of general linear models. Why? What happened to the random effects? Surely it would be better to interpret the coefficients from the mixed-effects models, which are likely more robust?

*Lines 498–500: What is the hypothesis here? That CTIr is the same across all levels of the predictor? If so, what levels vary from what other levels, and what is the importance of this insight? Also, given the massive differences in sample sizes among levels for most groups, it seems unlikely that assumptions of homoscedasticity of variances is met. As mentioned previously, if these numbers are based on counts, it might be better to explore some form of Poisson GLM (or better still, GLMM—so that the random effects can be included).

Minor editorial comments:

Overall, the manuscript was well constructed, but it would benefit from a thorough copy edit. Some of the suggestions below are a matter of taste, but others reflect a lack of consistency in writing.

Line 1 (and elsewhere): “cross-taxon”, not “cross-taxa”; adjectives like this are singular, not plural. A good example is available at the front of your title.

Line 65 (and elsewhere): add “concentrations” after “oxygen”.

Lines 75–76: Insert commas: “...semi-enclosed basins, such as the Mediterranean and Baltic

Seas, where physical...”

Lines 106–107: I suggest a slight revision to the sentence “All European Seas HAVE BEEN warming since AT LEAST the 1980’s, but WITH faster RATES in the semi-enclosed basins...”

Lines 112–114: This sentence is tricky. Time series cannot “correspond” to species. And it is unclear whether all time series cover the last 4 decades or whether they are simply located within this timeframe (the same issue exists in the Abstract). Consider revising for clarity.

Line 119: The phrase “European Seas basins” reads awkwardly because of the pluralisation of “Seas”. Perhaps “...basins within semi-enclosed European Seas...”?

Line 121: “...than IS THE CASE in the well-connected...”

Line 123: The concept of a “hotspot of high biodiversity climate velocity” eludes me.

Wording requires some thought here.

Line 130 and elsewhere: “i.e.” and “e.g.” should preferably be followed by a comma, but as it stands, this notion is inconsistently applied, throughout.

Lines 132–134: Are the signals or the communities associated with warming? As written, it seems to be the latter?

Line 150: The use of “their” seems strange, here.

Lines 171–172: Consider “...increased from 1980–2021, both AT the surface...and IN the water column...”

Lines 173: Consider “rate of ocean warming at the surface was...”. Also, note that SST and ST100 are undefined abbreviations at this point in the MS. I suspect that you have defined these in the Methods, but the arrangement of this MS means that the reader might not have got there, yet.

Line 186: GODAS is undefined at this point.

Line 188: Do you mean “...rate of change in CTI...?”

Line 189: Consider replacing “their” with “corresponding”. Also, “site labelling”, not “sites labelling”

Lines 191–192: This wording seems to suggest that year was included as both a random and a fixed effect? This needs clarification.

Lines 198–199: Why a mix in singular and plural descriptors in parentheses?

Lines 202–203: “...weaker trends...”? I might be better to frame these as “...trends statistically indistinguishable from zero...”?

Lines 214–215: Table 1 – what are the asterisks? It would also be good to explain the

bolding.

Line 226: Most frequent on a per-species basis? Or per location? Or some weighting?

Line 245: “whether”, not “if”

Lines 262–271: This material might better suit a summary table?

Lines 255–416: I read these, but cannot assess the robustness of interpretation until questions around the analyses are resolved.

Line 425: “...FOR the PERIOD 1980–2020...”

Line 426: “on” not “to”

Lines 449–450: What is meant by “A, the species’ abundance divided by the abundances of all species”? Do you mean “A, the species’ abundance divided by the total number of individuals at a site”? Would it not be simpler to refer to this as “relative abundance” or similar?

Lines 460–461: Delete “each”.

Line 463: A “temporal trend” (= change in a variable per unit time) is by definition a “rate”.

Lines 463–471: After having read the paragraph remain unclear about what is meant by “independently” and “jointly”.

Lines 467–468: The meaning of “CTI_r mean change was tested using: 1) a t-test weighted by the number of species present in each site...” is unclear to me. Statistical tests require an a priori hypothesis. What is the hypothesis about the trend being tested? Is this a t-test of the hypothesis that the slope of the relationship is zero?

Line 486: Consider “Analysis across regions and biotic groups”

Line 494: “whether” not “if”

Lines 500–502: I might have missed it, but I didn’t see this in the Results?

Reviewer #2 (Remarks to the Author):

This manuscript calculates CTI change in European seas and decomposes CTI to detect the relative influence of deborealization vs tropicalization in long-term monitoring datasets. This latter element is the most exciting contribution, given the extensive literature that already exists on changing CTI (Burrows et al. 2019), species richness (Blowes et al. 2019), abundance change relative to range position (Hastings et al. 2020), and community composition (Rutterford et al. 2023)—to name a few—in this well-studied region. (Given

that all these papers, much like the present manuscript, aim to indirectly gain inference into ecological processes by analyzing long-term surveys, I was surprised to read L92-94 in this manuscript which states that such studies are “scarce”.)

The manuscript is well-written and I agree that its goals—trying to understand the degree to which tropicalization vs. deborealization are driving community change in warming seas—are important. However, I have several issues that make it hard to assess the results of the paper as is. The first is the conflation of tropicalization with range expansion and deborealization with range contraction, which is captured in Fig. 1 and motivates much of the analysis. This assumes that the abundant-center hypothesis is true, which has very limited empirical support (Dallas et al. 2020) and the authors do not test for the species studied. For this reason, I was very confused by the attempt to back out retraction and expansion processes from CTI change (L498-502). The methods describing this are brief and not referenced (and it’s not clear if this is expansion/retraction of populations, or the whole range, in which case the authors should ensure that they are actually capturing range edges in the surveys), so maybe there is a very clear rationale that I just don’t know. Without that rationale, my understanding is: when we see tropicalization and deborealization, it is just that—abundance change of species with relatively higher or lower thermal affinities—and not necessarily range change unless we actually go measure range shifts.

Relatedly, while I understand the authors may be up against a limit to references, I encourage you to unpack and support the hypotheses a bit more (e.g., L143-148). I disagree that tropicalization and borealization processes are “poorly understood”; there are lots of large-scale studies and predictions to draw from here. For example, it’s possible that cold range edges of marine fishes are tracking temperature better than warm edges (a long-standing prediction of terrestrial biogeography theory), which would predominantly lead to tropicalization (Fredston-Hermann et al. 2020). On the other hand, some fish habitat models project faster climate velocities at trailing edges than at leading edges (Robinson et al. 2015), and some terrestrial species have even greater “colonization credit” than they have “extinction debt” (Talluto et al. 2017), leading to range contraction and therefore suggesting predominant deborealization.

My second concern is that there are extensive deficiencies in this manuscript's reporting summary and data/code availability relative to the Nature Portfolio standards and our field's best practices for open and transparent science which make it challenging to assess the methods and impossible to reproduce them. Unless I missed it, sample sizes are not reported in the main text. The reporting summary points to Supp. Tab. 1 but I can't find sample sizes there either. They should be reported as the number of records actually analyzed, not just number of species per dataset, and broken down by system and clade so readers can understand which groups and surveys drove the results. The code and raw data are also missing. I understand that not all the data are public but at a minimum the authors should share all of the code as well as the subset of the raw and processed datasets that they are able to publish. A data availability statement is also missing from the manuscript. In the reporting summary section on code availability, the authors pasted a citation for R rather than access information for their code. In the reporting summary section on data availability, the authors stated that raw data for some surveys are unavailable and that the processed data is described in a table, rather than providing access information for the raw data that are available.

I have a number of questions about the data analysis and statistics—which are really the heart of this paper—which are not answered in the brief section summarizing them (L452-471). Were the OBIS records filtered or quality-checked at all? How exactly were they merged with GODAS (the text says “annual means”—is that the year of the OBIS record?) Were models fitted to individual surveys or were surveys pooled? If the former, how did the authors correct for multiple testing and confirm that each individual time-series was not underpowered? How precisely was partial autocorrelation “checked for each yearly data time series”? Where are the model results reported and how many models were there? What were the two groups being compared in the t-test? Did the authors analyze the effect of sampling intensity / time-series length / sample size on the significance of each individual CTI trend?

Finally, I'm not convinced that this post-hoc ANOVA approach and the pie and violin charts in Fig. 3-4 are the best way to explore which species drove CTI change. More plots of the raw data, like a main text figure showing species' thermal bias (thermal affinity – CTI) vs.

abundance change, would really help to show this visually. A single species' influence on CTI is a combination of its abundance (since CTI in this analysis is abundance-weighted, I think) and the magnitude of its thermal bias. Ranking species by their influence on the CTI score and then reporting whether the changes in the most influential species are consistent with deborealization / tropicalization / etc and whether abundance change or differential thermal bias seem to be most important would keep the statistics "closer to the data" than doing an ANOVA on coefficients from a regression, and would enable the authors to unpack the underlying processes they focus on more than the current methods allow. I'm sure there are lots of other ways to think about this but I definitely encourage the authors to take a step back, connect the statistical approach with the processes they want to measure, focus more on effect sizes and which species are driving results, and less on p-values.

Because of these fundamental questions, I haven't really reviewed the results and discussion here. As I said earlier, I think this is an important research direction and an exciting dataset, and I would be happy to review a revised manuscript that resolves these methodological issues. I don't think these issues are insurmountable: expanding on the methods, sharing the data and code, and refocusing the text and results on how shifts in abundance of different species are driving different patterns of community turnover in these regions will turn this into a great paper.

Blowes SA, Supp SR, Antão LH, Bates A, Bruelheide H, et al. 2019. The geography of biodiversity change in marine and terrestrial assemblages. *Science*. 366(6463):339–45

Burrows MT, Bates AE, Costello MJ, Edwards M, Edgar GJ, et al. 2019. Ocean community warming responses explained by thermal affinities and temperature gradients. *Nat. Clim. Change*. 9(12):959–63

Dallas TA, Santini L, Decker R, Hastings A. 2020. Weighing the Evidence for the Abundant-Center Hypothesis. *Biodivers. Inform.* 15(3):81–91

Fredston-Hermann A, Selden R, Pinsky M, Gaines SD, Halpern BS. 2020. Cold range edges of marine fishes track climate change better than warm edges. *Glob. Change Biol.* 26(5):2908–22

Hastings RA, Rutterford LA, Freer JJ, Collins RA, Simpson SD, Genner MJ. 2020. Climate

Change Drives Poleward Increases and Equatorward Declines in Marine Species. *Curr. Biol.* 30:1572–77

Robinson LM, Hobday AJ, Possingham HP, Richardson AJ. 2015. Trailing edges projected to move faster than leading edges for large pelagic fish habitats under climate change. *Deep Sea Res. Part II Top. Stud. Oceanogr.* 113:225–34

Rutterford LA, Simpson SD, Bogstad B, Devine JA, Genner MJ. 2023. Sea temperature is the primary driver of recent and predicted fish community structure across Northeast Atlantic shelf seas. *Glob. Change Biol.* 29(9):2510–21

Talluto MV, Boulangeat I, Vissault S, Thuiller W, Gravel D. 2017. Extinction debt and colonization credit delay range shifts of eastern North American trees. *Nat. Ecol. Evol.* 1(7):1–6

Author Response (AR) to REVIEWER COMMENTS

Reference: NCOMMS-23-14566A

Notes: Author Response (AR) is indicated in blue. Lines correspond to the version without track changes.

Reviewer #1 (Remarks to the Author):

This is a neat paper that attempts to address an important ecological phenomenon. I particularly like the authors' intent to explore CTI change in the light its constituent components (topicalisation, demoralisation, etc.). As such, I would really like to see this work published. However, to properly assess the robustness of its findings, more detail about the analytical approaches is required—and depending on the outcome of that process, potentially also reanalysis of the data.

I have several major comments, as well as minor editorial suggestions.

Major comments:

1 – There were several unanswered questions about how the various groupings were constituted and what the group names really meant. Since this is a central focus of the manuscript, more detail/rationale is required here. Line-by-line examples follow.

*Line 183: Fig. 1b. Were crustaceans and molluscs included among the benthos, if not, why not (I don't see an answer in the Methods)?

*Lines 199–202: These results raise a further issue with separating molluscs and crustaceans from other benthos, unless the other benthos was sessile/burrowing, in which case, better names might be useful.

AR: The terminology employed was not clear as the reviewer pointed out. In benthos, there are crustaceans and molluscs among other macroinvertebrates, but these organisms were mainly sessile (in hard-bottom) or inhabiting the sediment and located in coastal areas (most of them at intertidal areas), while the groups of crustaceans and molluscs (now named cephalopods more precisely) were collected offshore using fishing or bottom trawl surveys, hence representing different benthic communities. We therefore renamed the biological groups and defined them in Methods section (lines 466-479):

1. Fish: pelagic, demersal and estuarine fish species.
2. Crustacea was renamed to demersal crustaceans: Demersal crustaceans collected offshore using bottom trawl fishing and surveys.
3. Mollusca was renamed to Cephalopods: Demersal and pelagic cephalopods collected offshore using bottom trawl and purse-seine fishing and surveys.
4. SB benthos was renamed to Coastal soft-bottom benthic macroinvertebrates: benthic macroinvertebrates species in coastal intertidal or subtidal soft-bottom sites, using grab or box corer samples.
5. HB benthos was renamed to Coastal hard-bottom benthos: benthic macroinvertebrates, macroalgae, lichens, and coralligenous assemblages in intertidal or subtidal, coastal, hard-bottom sites
6. Zooplankton: includes copepods and other zooplankton species collected by vertical net hauls

2 – There were several issues with the analyses. Although the authors demonstrate knowledge

of most of the issues (e.g., temporal [and to some degree, spatial] autocorrelation, the need for mixed models, etc.), the application of techniques was inconsistent, and in some cases, (given the information provided in the manuscript) questionable. The most important of these is the apparent abandonment of the results from the mixed-effects model to produce test statistics in the main results table (Table 1), rendering their interpretation open to questions about the actual alpha-level (probability of committing a type I error) used in the analyses. There are several other issues listed line-by-line, below. Some of these issues might be resolved by more careful explanation of the rationale and philosophy of the individual analyses (models), but in other cases, reconsidering the approaches to analyses might be warranted. Irrespective, it would be very useful to include diagnostic plots of the final fitted models in the Supplement, as well as a short section on caveats in the Discussion.

AR: In both temperature and CTI change over time analysis we now used linear mixed models (Zuur et al. 2009) consistently with sampling site as the random effect and year as the fixed effect, and adding a temporal autoregressive function (Mudelsee, 2019). The analysis of CTI_r by factors (biological group, basin type, habitat, region) (Table 1) was now undertaken using also consistent linear mixed models with temporal autoregressive function. Further we added a model selection analysis to identify the most representative factors underpinning CTI changes over time and addressing type I error, in particular, using the Akaike's Information Criterion corrected (AICc) by comparing all model combinations (Burnham & Anderson, 2002) (see Methods section, lines 504-516 and 547-558).

Diagnostic plot: As suggested, we performed diagnostic plots of residuals for the two selected factors (See new Supplementary Figure 3). Diagnostic plots did not show any major pattern, indicating that the model is reliable and adequate for this analysis.

Burnham, K. P., and D. R. Anderson. 2002. Model selection and multi-model inference: A practical information-theoretic approach. Springer.

Zuur, A. F., E. N. Ieno, and N. J. Walker. 2009. Mixed effects models and extensions in ecology with R. Springer Science, New York.

Mudelsee M. Trend analysis of climate time series: A review of methods. *Earth-Sci Rev* 190, 310-322 (2019).

*Lines 203–205: Given that there is only one observation from estuaries, would it not be more rigorous to simply omit that datum than to point out that a single point had the strongest trend, albeit non-significant? I am surprised that the algorithm managed to estimate a p-value from a single point...although this would have depended on the structure of the model, I guess. Nevertheless, the inclusion of this point is somewhat problematic.

AR: We included two new estuarine time series (“INRAE fish Pertuis” and “INRAE fish Basque”, see Supplementary Table 1) thanks to efforts in compiling new data and research collaboration. Now, three estuarine time series have been analysed with the improved and more consistent time series analysis. The two included estuarine time series showed also positive increase in CTI over time, as the first original one.

*Line 216–217: Fig. 3 – How were pies computed? On a per-species basis? How were these weighted when there were multiple locations/sites? This question further challenges the wisdom of including “estuary” as a habitat with only a single observation.

AR: Pie charts were computed estimating the percentage of the prevailing underlying ecological process ((de)tropicalization, and (de)borealization) for each case study and factor (Biological group, habitat, region, basin type). (see caption of Figure 3). As mentioned before, now there are three estuarine time series.

*Line 244–247: Table 2 and associated results – I’m not convinced that an ANOVA is the right way to go here. These are counts, so a log-linear analysis (saturated Poisson glm that includes interactions between grouping factors and process to test for independence of frequencies) seems more defensible...or perhaps a Poisson generalised mixed model that also includes the random effects?

AR: In Table 2, the variable “Tropicalization minus Deborealization” are not counts, it is a difference between percentages. Also, the underlying ecological processes (tropicalization, deborealization, ...) are computed for each site and over the entire time period, hence, each time series yield a single value of each underlying process. For these reasons, we compared the means of the underlying ecological processes intensities when CTI_r increased (in particular, tropicalization minus borealization) across different factors and their levels. To do so, we used linear models (Chambers, 1992) and added a selection of the best model using the AICc by comparing all factor combinations, or with forward stepwise selection in case of singularities among factors (Burnham & Anderson, 2002) (see section Methods lines 560-574). For comparison purposes, we also modelled the intensity of underlying processes at per-species basis with a linear mixed model for CTI_r>0 as a function of factors, with sites as random effect (lines 572-574); both analyses provide similar results (see new Supplementary Table 2, shown below).

Supplementary Table 2:

Factor	Level	n	CTI _r >0			
			Tropicalization minus Deborealization mean (°C y ⁻¹)	Tropicalization minus Deborealization SE	p-value	AICc
Biological group	CHB benthos	6	0.0233	0.0194	0.2342	-3577.94
	CSB benthos	18	0.0239	0.0115	0.0416	
	Zooplankton	3	-0.0158	0.0288	0.5853	
	Demersal crustaceans	4	0.0238	0.0207	0.2559	
	Cephalopods	5	-0.0031	0.0210	0.8822	
	Fish	16	-0.0200	0.0114	0.0838	
Habitat	Benthic / demersal	40	0.0175	0.0055	0.0023	-3623.56
	Estuarine	3	0.0043	0.0221	0.8458	
	Pelagic	8	-0.0908	0.0154	<0.0001	
Region	Baltic Sea	4	0.0043	0.0241	0.8582	-3596.88
	Mediterranean Sea	19	-0.0133	0.0100	0.1912	
	Northeast Atlantic	29	0.0198	0.0090	0.0321	
Basin type	Non-enclosed seas	26	0.0215	0.0094	0.0263	-3604.01
	Semi-enclosed seas	26	-0.0093	0.0088	0.2946	

Burnham, K. P., and D. R. Anderson. 2002. Model selection and multi-model inference: A practical information-theoretic approach. Springer.

Chambers J. M. 1992. Linear models. In: Statistical Models in S, edited by J. M. Chambers and T. J. Hastie. Wadsworth & Brooks/Cole.

*Lines 428–429: No consideration of temporal autocorrelation here? This needs to be addressed or at least listed as a caveat, with a brief mention of the likely consequences. You do this in Lines 465–467 for CTIr, so why not temperature?

AR: Accordingly with this comment and previous ones in relation to the consistency of analysis, we analysed the slope of the temperature trends and its significance using a linear mixed model (Zuur et al. 2009) of sea temperature with year as fixed effect, the sampling sites as random effect, and adding an autoregressive function to take into consideration the temporal autocorrelation (Mudelsee, 2019) (See Method section lines 551-559).

Zuur, A. F., E. N. Ieno, and N. J. Walker. 2009. Mixed effects models and extensions in ecology with R. Springer Science, New York.

Mudelsee M. Trend analysis of climate time series: A review of methods. *Earth-Sci Rev* 190, 310-322 (2019).

*Lines 433–439: A more detailed explanation for the soft-bottom/hard-bottom benthos vs mollusca/crustacea classifications are needed here, as is more detail on the durations of the time series (at least median, quartiles and range). Data are critical in this analysis, so such brief coverage is problematic.

AR: as mentioned above, we renamed the biological groups and defined in more detail in Methods section (lines 466-479):

1. Fish: pelagic, demersal and estuarine fish species.
2. Crustacea was named to demersal crustaceans: Demersal crustaceans collected offshore using bottom trawl fishing and surveys.
3. Mollusca was named Cephalopods: Demersal and pelagic cephalopods collected offshore using bottom trawl and purse-seine fishing and surveys.
4. SB benthos was named Coastal soft-bottom benthic macroinvertebrates: benthic macroinvertebrates species in coastal intertidal or subtidal soft-bottom sites, using grab or box corer samples.
5. HB benthos was named Coastal hard-bottom benthos: benthic macroinvertebrates, macroalgae, lichens, and coralligenous assemblages in intertidal or subtidal, coastal, hard-bottom sites.
6. Zooplankton: includes copepods and other zooplankton species collected by vertical net hauls.

The time series analysed here have varying durations over the last four decades (from 1980 to 2022 in the longest case). Most of time series span more than 15 years (92%), with median of 25 years, minimum of 9 years, and 20 and 39 years the 25% and 75% quartiles, respectively (see Methods section (lines 474-479)).

*Lines 443–461: Move all of the technical information into a single paragraph before explaining how the metric is used. Otherwise, the reader is left wondering how Ts and A are determined.

AR: This paragraph was rewritten accordingly (see lines 483-502).

*Lines 468–471: First, temporal autocorrelation seems to be missing from this analysis.

AR: Temporal autocorrelation has been now taken into consideration in the new linear mixed model analysis (see above and section Methods).

Second, the description here clashes with information in the caption for Figure 2: if site/taxonomic group is the random factor, how can Fig. 2c be showing random effects of the mixed model? For models of biological group, surely the random effect should be site nested within group (i.e., the effect of site doesn't disappear because you are more interested in taxa)?

AR: CTI change over time analysis was now based on linear mixed models (Zuur et al. 2009) consistently with sampling site as the random effect and year as the fixed effect, and adding a temporal autoregressive function (Mudelsee, 2019) (see Methods section, lines 547-558). In Fig. 2c, partial residuals of CTI overtime based on the mentioned model are showed (axis title clarified) and legend explanation was rewritten to: "partial residuals of CTI were calculated as CTI minus the random effect of sampling site of the linear mixed model with year as fixed effect and site as random effect", see lines 207 (Fig 2 caption). Hopefully, the figure is clearer now.

Finally, what is the effect of multiple tests, here? It is true that corrections for multiple tests are usually deployed for t-tests and the like, but the principle holds for any sequence of hypothesis tests. Here, you are refitting different models to the same response data. This increases the risk of a Type I error. Given the marginal significance of several of the results in Table 1, this becomes more problematic. One solution might be to fit a model with all predictors included, then simplify on the basis of AIC or some other information-theoretic criterion, and interpret only the predictors that remain? Irrespective, this is a serious issue here. Proper diagnostic plots of the final fitted models would also be useful. Finally, since the results depend on these analyses, a clearer and more detailed explanation of the underlying models (including their underlying rationale/philosophy) would really help to clarify what is being done.

AR: As mentioned above, we now used linear mixed models (Zuur et al. 2009) to analyse CTI change consistently with sampling site as the random effect and year as the fixed effect, and adding a temporal autoregressive function (Mudelsee, 2019). The analysis of CTI by groups (Table 1) was now undertaken using also consistent linear mixed models with temporal autoregressive function, and we added model selection to identify the most representative factors and addressing type I error, in particular, using the Akaike's Information Criterion corrected (AICc) by comparing all model combinations (Burnham & Anderson, 2002) (see Methods section with more detailed explanations, lines 547-558).

As suggested, we performed diagnostic plots of residuals for the two selected factors of the linear mixed model (See new Supplementary Figure 3, showed below). Diagnostic plots did not show any major pattern, indicating that the model is reliable and adequate.

Supplementary Figure 3

References:

Burnham, K. P., and D. R. Anderson. 2002. Model selection and multi-model inference: A practical information-theoretic approach. Springer.

Zuur, A. F., E. N. Ieno, and N. J. Walker. 2009. Mixed effects models and extensions in ecology with R. Springer Science, New York.

Mudelsee M. Trend analysis of climate time series: A review of methods. Earth-Sci Rev 190, 310-322 (2019).

*Lines 475–484: This does not provide any real insight into how the frequency(?) pies were computed for presentation in Figures 2 & 3. Again, without understanding what these plots actually represent, it is impossible to properly assess their meaning.

AR: For Fig 2d pie chart, we computed the percentages of the prevailing underlying process ((de)tropicalization, and (de)borealization) over all biodiversity time series (text added in lines

217-219, Fig 2 caption). Pie charts of Figure 3 were computed estimating the percentage of the prevailing underlying process ((de)tropicalization, and (de)borealization) in each case study and by factor (Biological group, habitat, region, basin type) (see legend of Figure 3).

*Lines 493–496: Here, it seems as if the mixed-effects models are simply ignored in favour of general linear models. Why? What happened to the random effects? Surely it would be better to interpret the coefficients from the mixed-effects models, which are likely more robust?

AR: As mentioned above, we now used linear mixed models (Zuur et al. 2009) to analyse CTI change consistently with sampling site as the random effect and year as the fixed effect, and adding a temporal autoregressive function (Mudelsee, 2019). The analysis of CTI_r by groups (Table 1) was now undertaken using also consistent linear mixed models with temporal autoregressive function, and we added model selection to identify the most representative factors and addressing type I error, in particular, using the Akaike's Information Criterion corrected (AIC_c) by comparing all model combinations (Burnham & Anderson, 2002) (see Methods section, lines 547-558).

*Lines 498–500: What is the hypothesis here? That CTI_r is the same across all levels of the predictor? If so, what levels vary from what other levels, and what is the importance of this insight? Also, given the massive differences in sample sizes among levels for most groups, it seems unlikely that assumptions of homoscedasticity of variances is met. As mentioned previously, if these numbers are based on counts, it might be better to explore some form of Poisson GLM (or better still, GLMM—so that the random effects can be included).

AR: As positive CTI changes correspond to an increased prevalence in warm-affinity species (i.e., tropicalization), and/or a decrease in cold-affinity species (i.e., deborealization), here, the aim is to test if tropicalization minus deborealization varies across levels of the different factors (text now added in Methods lines 560-565). As mentioned above, the variable “Tropicalization minus Deborealization” are not counts, it is a difference between percentages. Also, the underlying processes (tropicalization, deborealization, ...) are computed for each site and over the time period, hence, each time series yield a single value for each underlying process. For these reasons, the analysis was based on the comparison of the means of intensity of underlying processes (tropicalization minus borealization), for the set of time series where CTI_r increased, across different factors and their levels using linear models (Chambers, 1992), and we selected the best model using the AIC_c by comparing all factor combinations, or with forward stepwise selection in case of singularities among factors (Burnham & Anderson, 2002) (see section Methods lines 560-574).

Concerning homoscedasticity of variances, we undertook a Levene test, all groups were not significant ($p > 0.05$) indicating equal variances, except Basin group with $p = 0.0217$. However, we performed a non-parametric test (Kruskal Wallis test) that indicates, in agreement with the linear model and AIC_c, that Basin type is the most significant factor ($p < 0.0001$) and that tropicalization prevailed deborealization in non-enclosed seas.

References:

Burnham, K. P., and D. R. Anderson. 2002. Model selection and multi-model inference: A practical information-theoretic approach. Springer.

Chambers J. M. 1992. Linear models. In: Statistical Models in S, edited by J. M. Chambers and T. J. Hastie. Wadsworth & Brooks/Cole.

Minor editorial comments:

Overall, the manuscript was well constructed, but it would benefit from a thorough copy edit. Some of the suggestions below are a matter of taste, but others reflect a lack of consistency in writing.

Line 1 (and elsewhere): “cross-taxon”, not “cross-taxa”; adjectives like this are singular, not plural. A good example is available at the front of your title.

AR: corrected, “Cross-basins and cross-taxon patterns in biodiversity turnover in warming seas”

Line 65 (and elsewhere): add “concentrations” after “oxygen”.

AR: added

Lines 75–76: Insert commas: “...semi-enclosed basins, such as the Mediterranean and Baltic Seas, where physical...”

AR: added

Lines 106–107: I suggest a slight revision to the sentence “All European Seas HAVE BEEN warming since AT LEAST the 1980’s, but WITH faster RATES in the semi-enclosed basins...”

AR: corrected

Lines 112–114: This sentence is tricky. Time series cannot “correspond” to species. And it is unclear whether all time series cover the last 4 decades or whether they are simply located within this timeframe (the same issue exists in the Abstract). Consider revising for clarity.

AR: CTI was applied to long-term time series from 63 biodiversity programs spanning the last four decades (in the longest case), which accounts for 1806 species, including zooplankton, benthos, demersal and pelagic assemblages

Line 119: The phrase “European Seas basins” reads awkwardly because of the pluralisation of “Seas”. Perhaps “...basins within semi-enclosed European Seas...”?

AR: rewritten: “semi-enclosed sea basins with lower ocean connectivity to warmer waters are hypothesized to experience less tropicalization compared to deborealization than is the case in the well-connected northeast Atlantic region”

Line 121: “...than IS THE CASE in the well-connected...”

AR: corrected, see previous comment

Line 123: The concept of a “hotspot of high biodiversity climate velocity” eludes me. Wording requires some thought here.

AR: we rewrote the sentence to clarify “biodiversity hotspots associated to climate velocity”

Line 130 and elsewhere: “i.e.” and “e.g.” should preferably be followed by a comma, but as it stands, this notion is inconsistently applied, throughout.

AR: revised commas and use of terms; e.g. as “for example” and i.e. as “that is”

Lines 132–134: Are the signals or the communities associated with warming? As written, it seems to be the latter?

AR: revised: “Long-term monitoring programs based on permanent stations have detected such changes in North Atlantic fish communities, based on temporal changes of the CTI”

Line 150: The use of “their” seems strange, here.

AR: changed to “its”

Lines 171–172: Consider “...increased from 1980–2021, both AT the surface...and IN the water column...”

AR: corrected

Lines 173: Consider “rate of ocean warming at the surface was...”. Also, note that SST and ST100 are undefined abbreviations at this point in the MS. I suspect that you have defined these in the Methods, but the arrangement of this MS means that the reader might not have got there, yet.

AR: changed and defined here

Line 186: GODAS is undefined at this point.

AR: defined

Line 188: Do you mean “...rate of change in CTI...?”

AR: corrected

Line 189: Consider replacing “their” with “corresponding”. Also, “site labelling”, not “sites labelling”

AR: corrected

Lines 191–192: This wording seems to suggest that year was included as both a random and a fixed effect? This needs clarification.

AR: The linear mixed model includes year as fixed effect and site as random effect. We rewrote the sentence to clarify this as follows: “Partial residuals of CTI across time, calculated as CTI minus the random effect of sampling site of the mixed model, see Methods and Fig 2 caption” (lines 207-2018)

Lines 198–199: Why a mix in singular and plural descriptors in parentheses?

AR: corrected, all in singular

Lines 202–203: “...weaker trends...”? I might be better to frame these as “...trends statistically indistinguishable from zero...”?

AR: changed

Lines 214–215: Table 1 – what are the asterisks? It would also be good to explain the bolding.

AR: asterisks are removed, bolding (significant values, $p < 0.05$) is indicated now.

Line 226: Most frequent on a per-species basis? Or per location? Or some weighting?

AR: on a per-site basis, now indicated

Line 245: “whether”, not “if”

AR: changed

Lines 262–271: This material might better suit a summary table?

AR: This information covers some of our results and it is compared with specific studies of sea warming

Lines 255–416: I read these, but cannot assess the robustness of interpretation until questions around the analyses are resolved.

AR: Discussion has now been slightly changed according to the new results

Line 425: “...FOR the PERIOD 1980–2020...”

AR: corrected

Line 426: “on” not “to”

AR: corrected

Lines 449–450: What is meant by “A, the species’ abundance divided by the abundances of all species”? Do you mean “A, the species’ abundance divided by the total number of individuals at a site”? Would it not be simpler to refer to this as “relative abundance” or similar?

AR: changed: “where n is the number of species in the community, T_s is the temperature preference of each species (s) and A_s is the relative abundance of species s (*i.e.*, the abundance of species s divided by the total number of individuals in the community at a site)”

Lines 460–461: Delete “each”.

AR: deleted

Line 463: A “temporal trend” (= change in a variable per unit time) is by definition a “rate”.

AR: “rate” deleted

Lines 463–471: After having read the paragraph remain unclear about what is meant by “independently” and “jointly”.

AR: hopefully now is clarified: “Temporal trend of CTI were estimated: 1) for each site independently, and 2) jointly for all sites to provide an overall estimate of CTI change.”

Lines 467–468: The meaning of “CTI_r mean change was tested using: 1) a t-test weighted by the number of species present in each site...” is unclear to me. Statistical tests require an a priori hypothesis. What is the hypothesis about the trend being tested? Is this a t-test of the hypothesis that the slope of the relationship is zero?

AR: We have rewritten the whole paragraph to clarify the methods:
“Temporal trends of CTI were estimated using two approaches: 1) for each site independently, and 2) jointly for all sites to provide an overall estimate of CTI change. 1) We estimated the CTI_r for each site. To avoid potential biases in the estimation of CTI_r due to temporal autocorrelation, partial autocorrelation has been checked for each yearly data time series. Subsequently, CTI_r was estimated fitting linear models using generalized least squares, and adding an autoregressive function when autocorrelation was detected. 2) We tested whether the CTI change for all sites on average is different from zero using two statistical methods: a) a t-test weighted by the number of species present in each site, and b) a fit of a linear mixed model of CTI with year as fixed effect and sampling sites as random effect. Correlation between CTI_r and sea temperature (surface and 100 m integrated column) at each site were tested using linear mixed model with biological group as random effect.”

Line 486: Consider “Analysis across regions and biotic groups”

AR: changed

Line 494: “whether” not “if”

AR: changed

Lines 500–502: I might have missed it, but I didn’t see this in the Results?

AR: shown in new Figure 5

Reviewer #2 (Remarks to the Author):

This manuscript calculates CTI change in European seas and decomposes CTI to detect the relative influence of deborealization vs tropicalization in long-term monitoring datasets. This latter element is the most exciting contribution, given the extensive literature that already exists on changing CTI (Burrows et al. 2019), species richness (Blowes et al. 2019), abundance change relative to range position (Hastings et al. 2020), and community composition (Rutterford et al. 2023)—to name a few—in this well-studied region. (Given that all these papers, much like the present manuscript, aim to indirectly gain inference into ecological processes by analyzing long-term surveys, I was surprised to read L92-94 in this manuscript which states that such studies are “scarce”.)

AR: Thanks for the references. Those references not cited in the original version (Blowes et al. 2019, Hastings et al. 2020) were now added (together with others (Pinsky et al. 2019, Smale et al. 2019) in the new version to improve the context and discussion. We agree that substantial number of studies exist exploring the ecological processes explaining community response to climate change. We are specifically interested in explaining deborealization and tropicalization, which is not studied so extensively and comparatively. We changed the sentence: “Although

our knowledge of marine species responses to climate change is substantial, at least in the North Atlantic Ocean (Philippart et al. 2011, Peck and Pinnegar 2018), comparative assessments to identify the underlying ecological processes (McLean et al. 2021) associated with marine community expansion, retraction and dispersal constraints are limited to unevenly distribution of monitoring programs (Poloczanska et al. 2016, Bowler et al. 2017, Blowes et al. 2019, Burrows et al. 2019, Burrows et al. 2020)".

References:

Pinsky ML, Eikeset AM, McCauley DJ, Payne JL, Sunday JM. Greater vulnerability to warming of marine versus terrestrial ectotherms. *Nature* 569, 108-111 (2019).

Smale DA, et al. Marine heatwaves threaten global biodiversity and the provision of ecosystem services. *Nature Climate Change* 9, 306-312 (2019).

The manuscript is well-written and I agree that its goals—trying to understand the degree to which tropicalization vs. deborealization are driving community change in warming seas—are important. However, I have several issues that make it hard to assess the results of the paper as is. The first is the conflation of tropicalization with range expansion and deborealization with range contraction, which is captured in Fig. 1 and motivates much of the analysis. This assumes that the abundant-center hypothesis is true, which has very limited empirical support (Dallas et al. 2020) and the authors do not test for the species studied. For this reason, I was very confused by the attempt to back out retraction and expansion processes from CTI change (L498-502). The methods describing this are brief and not referenced (and it's not clear if this is expansion/retraction of populations, or the whole range, in which case the authors should ensure that they are actually capturing range edges in the surveys), so maybe there is a very clear rationale that I just don't know. Without that rationale, my understanding is: when we see tropicalization and deborealization, it is just that—abundance change of species with relatively higher or lower thermal affinities—and not necessarily range change unless we actually go measure range shifts.

AR: We agree with the reviewer on that we are not actually testing geographical range change, and focusing our analysis in abundance change of species relative to their thermal affinities. Now we changed the use of "expansion" term to simply "tropicalization + borealization" and "retraction" to "detropicalization + deborealization". We also expanded the hypothesis in Methods (lines 560-574): "As positive CTI changes correspond to an increased prevalence in warm-affinity species (i.e., tropicalization), and/or a decrease in cold-affinity species (i.e., deborealization), the aim here is to test if tropicalization minus deborealization varies across levels of the different factors. The reason why tropicalization may prevail (or not) over deborealization might depend on biological traits (size, dispersal capacity), and seascape aspects (e.g., marine geomorphological features) limiting species dispersal and ocean connectivity, which has been scarcely tested with observations. For positive CTI_r, we hence tested different factors. [...] It can be also expected that seas with limited connectivity to open oceans would constrain species abundance increases (i.e., tropicalization and borealization) over abundance decreases (i.e., detropicalization and deborealization) processes in both CTI increase or decrease. Hence, for basin type, we tested whether (tropicalization + borealization) *minus* (detropicalization + deborealization) processes varied across levels for all dataset (i.e. CTI_r positive and negative)".

Relatedly, while I understand the authors may be up against a limit to references, I encourage you to unpack and support the hypotheses a bit more (e.g., L143-148). I disagree that tropicalization and borealization processes are "poorly understood"; there are lots of large-

scale studies and predictions to draw from here. For example, it's possible that cold range edges of marine fishes are tracking temperature better than warm edges (a long-standing prediction of terrestrial biogeography theory), which would predominantly lead to tropicalization (Fredston-Hermann et al. 2020). On the other hand, some fish habitat models project faster climate velocities at trailing edges than at leading edges (Robinson et al. 2015), and some terrestrial species have even greater "colonization credit" than they have "extinction debt" (Talluto et al. 2017), leading to range contraction and therefore suggesting predominant deborealization.

AR: We agree on the comments and accordingly we expanded and rephrased this paragraph: "The reason why tropicalization may prevail (or not) over deborealization has been recently debated (Robinson et al. 2015, Talluto et al. 2017, Fredston-Hermann et al. 2020). Assuming an ecophysiological equilibrium between habitat suitability and the occurrence of species, we can expect that marine species track temperature change equally (Fredston-Hermann et al. 2020; Sunday et al. 2012), hence tropicalization would be similar to deborealization. However, in species with slow demography and limited dispersal, lags between climate change and distribution shifts can result in 'extinction debts' (Jablonski 2001), where populations temporarily persist under unsuitable conditions, and 'colonization credits', where suitable locations are not occupied (Talluto et al. 2017). This latter pattern has been observed for instance in trees (Talluto et al. 2017), whilst other studies found equally responsive shifts at both range boundaries in marine ectotherms (Sunday et al. 2012), and prevalence in tropicalization in demersal fish (McLean et al. 2021). The prevalence in tropicalization vs deborealization might depend on biological traits of the community (size, dispersal capacity), and also seascape aspects (e.g., marine geomorphological features) limiting species dispersal and ocean connectivity. Thus, it can be expected that seas with limited connectivity to open oceans would constrain species abundance increases (tropicalization and borealization) over species abundance decreases (detropicalization and deborealization)."

My second concern is that there are extensive deficiencies in this manuscript's reporting summary and data/code availability relative to the Nature Portfolio standards and our field's best practices for open and transparent science which make it challenging to assess the methods and impossible to reproduce them. Unless I missed it, sample sizes are not reported in the main text. The reporting summary points to Supp. Tab. 1 but I can't find sample sizes there either. They should be reported as the number of records actually analyzed, not just number of species per dataset, and broken down by system and clade so readers can understand which groups and surveys drove the results. The code and raw data are also missing. I understand that not all the data are public but at a minimum the authors should share all of the code as well as the subset of the raw and processed datasets that they are able to publish. A data availability statement is also missing from the manuscript. In the reporting summary section on code availability, the authors pasted a citation for R rather than access information for their code. In the reporting summary section on data availability, the authors stated that raw data for some surveys are unavailable and that the processed data is described in a table, rather than providing access information for the raw data that are available.

AR: All codes are available in a public repository (<https://zenodo.org/records/10149019>, DOI: 10.5281/zenodo.10149018). Concerning data availability, Data generated during the study and that support their findings (all CTI time series, i.e. CTI per year for each site, and all underlying process scores ((de)tropicalization, (de)borealization) at per-species and per-site basis) are available in a public repository (<https://zenodo.org/records/10149019>, DOI: 10.5281/zenodo.10149018), and in Supplementary Table 1. Biodiversity original data (i.e., species abundance at each year for each site survey) is subject to restrictions as it pertains to

the corresponding institution. Certain data is, however, available from the corresponding author upon request.

Sample size encompasses the number of species to calculate CTI, and time period span to estimate CTI change. With the new analysis, this means that 65 sites account for 1730 “samples”. In most of datasets, the abundance is estimated with number of individuals, but in some datasets the abundance is determined by semi-quantitative sampling of surface coverage similar to Braun-Blanquet. However, the reader can now check species list in available dataset.

I have a number of questions about the data analysis and statistics—which are really the heart of this paper—which are not answered in the brief section summarizing them (L452-471). Were the OBIS records filtered or quality-checked at all? How exactly were they merged with GODAS (the text says “annual means”—is that the year of the OBIS record?) Were models fitted to individual surveys or were surveys pooled? If the former, how did the authors correct for multiple testing and confirm that each individual time-series was not underpowered? How precisely was partial autocorrelation “checked for each yearly data time series”? Where are the model results reported and how many models were there? What were the two groups being compared in the t-test? Did the authors analyze the effect of sampling intensity / time-series length / sample size on the significance of each individual CTI trend?

AR: OBIS records were quality-checked removing duplicates. To characterize the thermal preferences of the species local temperatures derived from GODAS were used, which was available from OBIS for each observation. This information has been explicitly included (line 490-502).

AR: As explained in previous comments, we now used linear mixed models (Zuur et al. 2009) to analyse CTI change consistently with sampling site as the random effect (i.e. all surveys analysed together) and year as the fixed effect, and including a temporal autoregressive function (Mudelsee, 2019), and adding model selection to identify the most representative factors and addressing type I error using the Akaike’s Information Criterion corrected (AICc) by comparing all model combinations (Burnham & Anderson, 2002) (see Methods section, lines 504-516 and 547-558). In this way, we avoid multiple testing now.

Finally, I’m not convinced that this post-hoc ANOVA approach and the pie and violin charts in Fig. 3-4 are the best way to explore which species drove CTI change. More plots of the raw data, like a main text figure showing species’ thermal bias (thermal affinity – CTI) vs. abundance change, would really help to show this visually. A single species’ influence on CTI is a combination of its abundance (since CTI in this analysis is abundance-weighted, I think) and the magnitude of its thermal bias. Ranking species by their influence on the CTI score and then reporting whether the changes in the most influential species are consistent with deborealization / tropicalization / etc and whether abundance change or differential thermal bias seem to be most important would keep the statistics “closer to the data” than doing an ANOVA on coefficients from a regression, and would enable the authors to unpack the underlying processes they focus on more than the current methods allow. I’m sure there are lots of other ways to think about this but I definitely encourage the authors to take a step back, connect the statistical approach with the processes they want to measure, focus more on effect sizes and which species are driving results, and less on p-values.

AR: The analysis at per-species suggested by the reviewer is indeed interesting and confirmed the original analysis at per-site basis. We addressed this idea and plot species’ thermal bias (thermal affinity – CTI) vs species’ abundance change (see new Figure 4, showed below). The

extreme tropicalized and deborealized species have been indicated in Figure 4 and discussed in Discussion section (lines 383-401). From underlying processes calculated at species level, we can now calculate the intensity of process at overall and per-site basis in order to provide estimates of the process at community level and analysis the potential effect of factors. To do that, we compared the means of intensity of underlying processes of increased CTIr (in particular, tropicalization minus borealization) of marine communities across different factors and their levels using linear models and selected the best model using the AICc by comparing all combinations, or with forward stepwise selection in case of singularities caused from interaction among factors. For comparison purposes to check if packing or unpacking data (as suggested by the reviewer) yield similar or different results, we also modelled the intensity of underlying processes at per-species basis with a linear mixed model for CTIr>0 as a function of factors, with sites as random effect (lines 276, 572). As mentioned, both analyses provide similar results (see new Supplementary Table 2).

Figure 4

Because of these fundamental questions, I haven't really reviewed the results and discussion here. As I said earlier, I think this is an important research direction and an exciting dataset, and I would be happy to review a revised manuscript that resolves these methodological

issues. I don't think these issues are insurmountable: expanding on the methods, sharing the data and code, and refocusing the text and results on how shifts in abundance of different species are driving different patterns of community turnover in these regions will turn this into a great paper.

AR: We expanded and detailed the methods, shared codes, and modified the results and conclusions accordingly.

Blowes SA, Supp SR, Antão LH, Bates A, Bruelheide H, et al. 2019. The geography of biodiversity change in marine and terrestrial assemblages. *Science*. 366(6463):339–45

Burrows MT, Bates AE, Costello MJ, Edwards M, Edgar GJ, et al. 2019. Ocean community warming responses explained by thermal affinities and temperature gradients. *Nat. Clim. Change*. 9(12):959–63

Dallas TA, Santini L, Decker R, Hastings A. 2020. Weighing the Evidence for the Abundant-Center Hypothesis. *Biodivers. Inform.* 15(3):81–91

Fredston-Hermann A, Selden R, Pinsky M, Gaines SD, Halpern BS. 2020. Cold range edges of marine fishes track climate change better than warm edges. *Glob. Change Biol.* 26(5):2908–22

Hastings RA, Rutterford LA, Freer JJ, Collins RA, Simpson SD, Genner MJ. 2020. Climate Change Drives Poleward Increases and Equatorward Declines in Marine Species. *Curr. Biol.* 30:1572–77

Robinson LM, Hobday AJ, Possingham HP, Richardson AJ. 2015. Trailing edges projected to move faster than leading edges for large pelagic fish habitats under climate change. *Deep Sea Res. Part II Top. Stud. Oceanogr.* 113:225–34

Rutterford LA, Simpson SD, Bogstad B, Devine JA, Genner MJ. 2023. Sea temperature is the primary driver of recent and predicted fish community structure across Northeast Atlantic shelf seas. *Glob. Change Biol.* 29(9):2510–21

Talluto MV, Boulangeat I, Vissault S, Thuiller W, Gravel D. 2017. Extinction debt and colonization credit delay range shifts of eastern North American trees. *Nat. Ecol. Evol.* 1(7):1–6

REVIEWERS' COMMENTS

Reviewer #1 (Remarks to the Author):

The authors have done a good job in addressing my initial queries. I have only a few minor comments, listed below.

Title: "Cross-basin" singular, like, "cross-taxon".

Line 79: "...in the CONTEXT OF ocean warming...". Throughout the MS the tendency to compound nouns should be avoided; alternatively, please hyphenate correctly to avoid ambiguity.

Remainder of the MS: I will refrain from further language editing and leave this to the Editorial staff to deal with. In my opinion, many grammatical errors and complications remain or have been introduced in revision.

Lines 226–227: Contrary to the wording, the confidence intervals seem to be appropriately displayed?

Lines 279–299: These descriptions to me don't really match what the pie charts are showing. I guess it depends on what is meant by "equal" and "dominate". For example, in panel a, the only group with what looks like "equal(ish) intensities" to me is crustacea.

Figure 4: It's not clear to me what the crossed arrows in the middle of the plot are meant to illustrate. Please either explain in the caption or remove.

Line 371–373: Given information in Figure 4, dominance is ~60:40. Is that really "highly dominant"? This ratio is a lot lower than I would have expected.

Line 403: Figure 5b doesn't seem to show changes in abundance?

Overall, I believe that the work in this manuscript is robust, but the writing remains an issue,

making it difficult to read/understand in places.

Reviewer #2 (Remarks to the Author):

The manuscript is vastly improved and I think the new analysis is much more statistically robust. Thank you for including Figure 4, which addresses many of my questions (although note that the figure legend and figure don't seem to match--the figure has arrows that aren't explained in the legend, and the legend mentions extreme values that aren't labeled on the figure.) I have no additional comments.

RESPONSE TO REVIEWERS' COMMENTS

Reviewer #1 (Remarks to the Author):

The authors have done a good job in addressing my initial queries. I have only a few minor comments, listed below.

Title: “Cross-basin” singular, like, “cross-taxon”.

AR: done

Line 79: “...in the CONTEXT OF ocean warming...”. Throughout the MS the tendency to compound nouns should be avoided; alternatively, please hyphenate correctly to avoid ambiguity.

AR: done

Remainder of the MS: I will refrain from further language editing and leave this to the Editorial staff to deal with. In my opinion, many grammatical errors and complications remain or have been introduced in revision.

AR: I leave Editorial editing to deal with.

Lines 226–227: Contrary to the wording, the confidence intervals seem to be appropriately displayed?

AR: Corrected. Sentence removed.

Lines 279–299: These descriptions to me don't really match what the pie charts are showing. I guess it depends on what is meant by “equal” and “dominate”. For example, in panel a, the only group with what looks like “equal(ish) intensities” to me is crustacea.

AR: Reviewer is correct. Descriptions refer to the model analysis on the relative importance of tropicalization with respect to deborealization (shown in Table 2) which is now indicated instead of Figure 3 pie charts.

Figure 4: It's not clear to me what the crossed arrows in the middle of the plot are meant to illustrate. Please either explain in the caption or remove.

AR: For all species in all case studies, arrows represent the mean value of Species' thermal bias with respect to Species' abundance change for each underlying process (i.e. tropicalization, borealization, detropicalization, deborealization). Now included in Figure 4 caption.

Line 371–373: Given information in Figure 4, dominance is ~60:40. Is that really “highly dominant”? This ratio is a lot lower than I would have expected.

AR: The analysis at per-site basis (Fig 2d) and at per-species basis (Figure 4) agree in that tropicalization and deborealization dominates detropicalization and borealization, but proportions are not the same. The underlying ecological processes explaining the increase in CTI can generally be attributed to the prevalence of tropicalization or deborealization in most of sites (76.9% of sites), whilst detropicalization and borealization dominated at fewer sites (23.1%) (Figure 2d). We removed “highly”.

Line 403: Figure 5b doesn't seem to show changes in abundance?

AR: Figure 5b show Species' abundance increases (i.e., tropicalization+borealization) relative to species' abundance decreases (i.e., detropicalization+deborealization). This is now included in the text.

Overall, I believe that the work in this manuscript is robust, but the writing remains an issue, making it difficult to read/understand in places.

Reviewer #2 (Remarks to the Author):

The manuscript is vastly improved and I think the new analysis is much more statistically robust. Thank you for including Figure 4, which addresses many of my questions (although note that the figure legend and figure don't seem to match-- the figure has arrows that aren't explained in the legend, and the legend mentions extreme values that aren't labeled on the figure.) I have no additional comments.

AR: For all species in all case studies, arrows represent the mean value of Species' thermal bias with respect to Species' abundance change for each underlying process (i.e. tropicalization, borealization, detropicalization, deborealization). Now included in Figure 4 caption.

Extreme values are now labelled with species names in new Figure 4.